# Treacle's ability to form liquid-like phase condensates is essential for nucleolar fibrillar center assembly, efficient rRNA transcription and processing, and rRNA gene repair

Artem K Velichko[1,2,3], Nadezhda V Petrova[1], Dmitry A Deriglazov[1], Anastasia P Kovina[1], Artem V Luzhin[1,2], Eugene P Kazakov[4], Igor I Kireev[4], Sergey Razin[1,5], Omar L Kantidze[1]*†

[1]Department of Cellular Genomics, Institute of Gene Biology RAS, Moscow, Russian Federation; [2]Center for Precision Genome Editing and Genetic Technologies for Biomedicine, Institute of Gene Biology RAS, Moscow, Russian Federation; [3]Institute for Translational Medicine and Biotechnology, Sechenov First Moscow State Medical University, Moscow, Russian Federation; [4]A.N. Belozersky Institute of Physico-Chemical Biology, Lomonosov Moscow State University, Moscow, Russian Federation; [5]Biological Faculty, Lomonosov Moscow State University, Moscow, Russian Federation

*For correspondence: kantidze@gmail.com

Present address: †Quantori LLC, Cambridge, United States

**Competing interest:** The authors declare that no competing interests exist.

## eLife Assessment

This **important** study reveals that the nucleolar protein Treacle undergoes liquid-liquid phase separation in vitro and in vivo. It provides **convincing** evidence that the ability of Treacle to form phase-separated condensates is necessary for the proper formation of the fibrillar center of the nucleolus, rRNA transcription, and rDNA repair. These findings will be of interest to the communities studying biomolecular condensates, nucleolar organization, and ribosome biogenesis.

**Abstract** We investigated the role of the nucleolar protein Treacle in organizing and regulating the nucleolus in human cells. Our results support Treacle's ability to form liquid-like phase condensates through electrostatic interactions among molecules. The formation of these biomolecular condensates is crucial for segregating nucleolar fibrillar centers from the dense fibrillar component and ensuring high levels of ribosomal RNA (rRNA) gene transcription and accurate rRNA processing. Both the central and C-terminal domains of Treacle are required to form liquid-like condensates. The initiation of phase separation is attributed to the C-terminal domain. The central domain is characterized by repeated stretches of alternatively charged amino acid residues and is vital for condensate stability. Overexpression of mutant forms of Treacle that cannot form liquid-like phase condensates compromises the assembly of fibrillar centers, suppressing rRNA gene transcription and disrupting rRNA processing. These mutant forms also fail to recruit DNA topoisomerase II binding protein 1 (TOPBP1), suppressing the DNA damage response in the nucleolus.

## Introduction

Applying polymer chemistry principles to biological molecules has significantly broadened our understanding of the functions of membraneless compartments (*Mehta and Zhang, 2022*; *Mittag and Pappu, 2022*). Multiple weak, cooperative, and dynamic interactions underlie the assembly of these self-organized structures, termed biomolecular condensates. Mechanistically, liquid-liquid phase separation (LLPS)—a demixing process that yields a condensed protein-enriched phase and a dilute phase—can mediate the self-assembly of biomolecular condensates. In biological systems, phase separation is facilitated by a combination of multivalent interactions mediated by intrinsically disordered regions and site-specific interactions that drive percolation (*Banani et al., 2017*; *Shin and Brangwynne, 2017*; *Uversky, 2019*). Biomolecular condensates serve as functional hubs in diverse cellular processes such as transcription (*Sabari et al., 2020*), microtubule nucleation (*Woodruff et al., 2017*), and the adaptive stress response (*Franzmann and Alberti, 2019*).

The nucleolus is a large and complex subnuclear compartment that can also be considered a multi-component and multilayered biomolecular condensate (*Lafontaine et al., 2021*). It is formed around arrays of ribosomal gene repeats (rDNA), which are transcribed by RNA polymerase I (RNA Pol I) to produce ribosomal RNA (rRNA) (*Cerqueira and Lemos, 2019*). In the nucleolus, RNAs and hundreds of different proteins are segregated into three immiscible phases: the fibrillar center (FC), which is the RNA Pol I transcription factory; the dense fibrillar component (DFC), which is the intermediate layer rich in fibrillarin (FBL); and the granular component (GC), which is the outer layer rich in nucleophosmin (NPM1/B23). The mechanisms underlying DFC and GC formation through phase separation are relatively well understood. Studies have demonstrated that a ternary mixture comprising the GC protein NPM1, a monomeric version of the DFC protein FBL, and generic rRNA is sufficient to form an FBL-enriched inner phase that coexists with an NPM-enriched outer phase (*Feric et al., 2016*; *Mitrea et al., 2016*; *Mitrea et al., 2018*). In contrast, the mechanisms of FC formation and the role of LLPS in this process have received considerably less attention.

Previous studies on FC organization in living cells suggest that FC assembly may also occur via phase separation (*Falahati et al., 2016*; *Falahati and Wieschaus, 2017*). Several lines of evidence support this hypothesis. First, FC components migrate to the nucleolar periphery and aggregate into large structures known as nucleolar caps when RNA Pol I transcription is inhibited by actinomycin D (AMD) or genotoxic drugs (*Harding et al., 2015*; *Korsholm et al., 2019*; *Reuynolds et al., 1964*). This structural transformation is reminiscent of liquid droplet fusion. Second, high-speed molecular dynamics indicative of fluid-like behavior have been observed in nucleolar caps using single-molecule tracking of RNA Pol I and chromatin-bound upstream binding transcription factor (UBFT/UBF) (*Ide et al., 2020*). Third, both UBF and its partner Treacle ribosome biogenesis factor 1 (TCOF1/Treacle) demonstrated the ability to undergo condensation in vitro and in vivo, respectively (*Jaberi Lashkari et al., 2023*; *King et al., 2024*). Therefore, it is likely that FC assembly and organization rely on LLPS, similar to other nucleolar subcompartments. However, further studies are needed to elucidate the molecular determinants and biophysical properties underlying this process.

In this context, the nucleolar phosphoprotein Treacle is of particular interest. As previously mentioned, it directly interacts with UBF and RNA Pol I, co-localizing with them within the FC. In addition, Treacle exhibits both transcription-dependent and transcription-independent functions (*Gál et al., 2022*). Its role in ribosome biogenesis is well documented since it assists in the transcription and processing of rRNA (*Gonzales et al., 2005*; *Lin and Yeh, 2009*; *Valdez et al., 2004*). It is also a master regulator of the DNA damage response (DDR) in the nucleolus, mediating the recruitment of repair factors to rDNA damaged by I-PpoI-induced breaks, replication stress, or R-loop formation (*Ciccia et al., 2014*; *Korsholm et al., 2019*; *Larsen et al., 2014*; *Mooser et al., 2020*; *Velichko et al., 2019*; *Velichko et al., 2021*). Therefore, it acts as a nucleolar hub with both transcriptional and DNA repair functions.

The multifunctionality of Treacle may be related to its ability to engage in multivalent interactions and undergo phase separation because of the presence of extended intrinsically disordered regions. In addition, Treacle's partners in transcription (RNA Pol I, UBF) and DNA repair (DNA topoisomerase II binding protein 1 [TOPBP1]) also utilize phase separation as a functioning mechanism (*Frattini et al., 2021*; *Ide et al., 2020*; *King et al., 2024*). Transcription of rDNA can be regulated by phase separation of RNA Pol I, while TOPBP1 condensation can activate the ATR serine/threonine kinase (ATR) signaling pathway (*Frattini et al., 2021*; *Ide et al., 2020*).

Considering these observations, we conducted an in-depth study of Treacle's condensation characteristics and their impact on the structural organization and functional dynamics of nucleoli. We found that Treacle can form biomolecular condensates and characterized its structural features regulating this process. Through its ability to condense, Treacle promotes the formation of nucleolar FCs by recruiting and concentrating transcription factors at rDNA and separating FC components from the DFC. Collectively, this segregates rRNA synthesis and subsequent processing in distinct compartments of the nucleolus. Impairing Treacle's condensation ability results in the mixing of FC and DFC components, reducing the efficiency of both processes, equivalent to completed depleting endogenous Treacle. We also demonstrate that Treacle's phase separation is critical for its interaction with TOPBP1 and activation of the DDR in rDNA during genotoxic stress. Our findings reveal the role of Treacle not only as a structural scaffold for FCs but also as a nucleolar hub integrating the functions of rDNA transcription, rRNA processing, and preserving rDNA integrity.

## Results

### Treacle is a scaffold protein for nucleolar FCs and DFCs

First, we defined the role of Treacle in the structural organization of the nucleolus. We employed the CRISPR/Cas9 system to knock out the *TCOF1* gene encoding Treacle in HeLa cells to obtain a population of cells lacking Treacle signals (Treacle-negative cells; *Figure 1—figure supplement 1A*). Immunocytochemical staining for UBF and RNA Pol I subunit A (POLR1A/RPA194; FC markers), FBL (DFC marker), and B23 and nucleolin (NCL; GC markers) revealed that Treacle-positive cells exhibited a classical tripartite nucleolar structure, with Treacle co-localizing with UBF1 and RPA194 in FCs, FBL in DFCs, and B23 and NCL surrounding DFCs (*Figure 1A*). Treacle depletion resulted in the disappearance of distinct FC and DFC structures, causing RPA194, UBF, and FBL to diffuse throughout the nucleolus (*Figure 1B*; *Figure 1—figure supplement 1A*). This suggests a mixing of FC and DFC components (*Figure 1—figure supplement 1B*). The organization of the GC component, however, was minimally affected (*Figure 1B*; *Figure 1—figure supplement 1C*). Notably, Treacle depletion also partially redistributed RPA194, UBF, FBL, and NCL from the nucleolus to the nucleoplasm, likely reflecting disrupted ribosomal gene transcription (*Figure 1B*; *Figure 1—figure supplement 1A*). Similar effects were observed in normal human skin fibroblasts following Treacle depletion (*Figure 1—figure supplement 1D and E*). These results indicated that Treacle plays a critical role in forming the integrated structure of FCs and DFCs.

Next, we investigated the impact of nucleolar FC disintegration on the efficiency of rDNA transcription and the recruitment of the RNA Pol I transcription machinery to rDNA. Chromatin immunoprecipitation (ChIP) analysis of cells transfected with an anti-Treacle single guide RNA (sgRNA) revealed a decreased occupancy of UBF and RPA194 within the rDNA coding region (*Figure 1C*). In contrast, transfection with a mock sgRNA did not cause such an effect (*Figure 1—figure supplement 2A*). These observations suggest that Treacle amplifies the recruitment rather than initiates the primary binding of the transcription machinery to rDNA. Depleting Treacle substantially suppressed rDNA transcription, although not completely, unlike the repression induced by AMD (*Figure 1D and E*; *Figure 1—figure supplement 2B*). Furthermore, cells depleted of Treacle exhibited impaired 5' external transcribed spacer processing, particularly in the cleavage of the A' site (*Figure 1F*), which is one of the initial steps in 18S rRNA maturation.

In summary, Treacle functions as a scaffold protein in nucleolar FCs, and its depletion compromises both rRNA gene transcription and processing.

### Treacle drives the formation of biomolecular condensates

Currently, it has become evident that various scaffold proteins can self-organize by forming liquid-phase condensates (*Banani et al., 2017*; *Shin and Brangwynne, 2017*). According to protein structure predictors (e.g. AlphaFold, IUPred2, PONDR, and FuzDrop), Treacle is a fully intrinsically disordered protein (*Figure 2—figure supplement 1A*). Nevertheless, Treacle can be broadly categorized into three distinct functional domains: the N-terminal domain (ND), the central domain, and the C-terminal domain (CD) (*Gál et al., 2022*). The ND of Treacle harbors a LisH motif and a nuclear localization signal (NLS). The central domain contains 15 low-complexity regions (LCRs) enriched in serine and glutamic acid residues (S/E-rich LCRs), which are interspersed with unstructured linker sequences

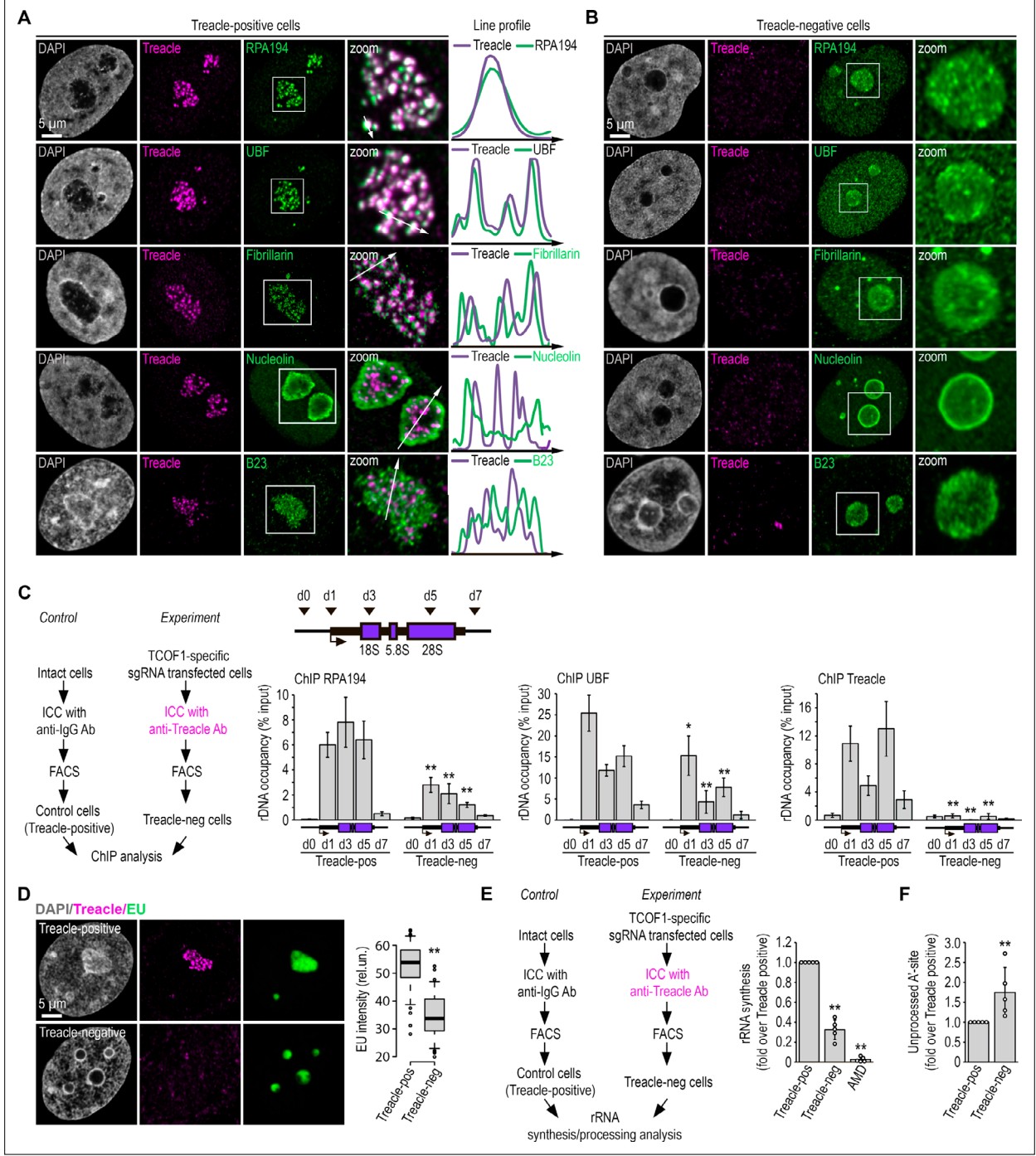

**Figure 1.** Treacle is a scaffold protein for nucleolar fibrillar centers (FCs) and dense fibrillar components (DFCs). (**A**) Intact HeLa cells (Treacle-positive) were fixed and co-immunostained with Treacle and with either RPA194, UBF, fibrillarin, B23, or nucleolin antibodies. DNA was stained with DAPI (gray). Cells were analyzed by laser scanning confocal microscopy. Representative images of cells and nucleoli (magnified images) are shown. Co-localization analysis was performed on the merged images. Graphs illustrate quantification in arbitrary units of fluorescence distribution along the lines shown in the figures. (**B**) HeLa cells were transfected with a construct coding CRISPR/Cas9 and single guide RNA (sgRNA) to the TCOF1 gene. 7–10 days after transfection, the cells were fixed and co-immunostained with Treacle and either RPA194, UBF, fibrillarin, B23, or nucleolin antibodies. DNA was stained with DAPI (gray). Cells were analyzed by laser scanning confocal microscopy. Representative images of Treacle-negative cells and nucleoli (magnified images) are shown. (**C**) HeLa cells were transfected with a construct coding CRISPR/Cas9 and sgRNA to the TCOF1 gene (*experiment*). 7–10 days after transfection cell were fixed, immunostained with Treacle antibodies, and subjected to cell sorting in the fluorescent analysis mode to obtain Treacle-negative populations. Intact HeLa cells (*control*) were fixed, immunostained with IgG antibodies, passed through all FACS-related procedure in the light scattering analysis mode, and used as a control (Treacle-positive). The sorted cell fractions were used for chromatin immunoprecipitation (ChIP)

*Figure 1 continued*

analysis with either Treacle, RPA194, or UBF antibodies. ChIP was followed by quantitative polymerase chain reaction (qPCR) using the d0, d1, d3, d5, d7 primers to the ribosomal RNA (rRNA) gene (positioned as indicated on the scheme). Data are represented relative to the input. Values are means ± SD from at least three independent replicates (**, p<0.01; *, p<0.1; by unpaired t test). (**D**) HeLa cells were transfected with a construct coding CRISPR/Cas9 and sgRNA to the TCOF1 gene. 7–10 days after transfection the cells were pulsed with EU (100 μM for 2 hr), fixed and immunostained with Treacle antibodies. EU (green) was revealed by click chemistry. The DNA was stained with DAPI (gray). Representative images of Treacle-positive and Treacle-negative HeLa cells are shown. Quantification of EU fluorescence intensities in Treacle-positive and Treacle-negative HeLa cells are shown in the right panel (**, p<0.01 by unpaired t test; n>500). (**E**) HeLa cells were transfected with a construct coding CRISPR/Cas9 and sgRNA to the TCOF1 gene (*experiment*). 7–10 days after transfection the cells were fixed, immunostained with Treacle antibodies, and subjected to cell sorting in the fluorescent analysis mode to obtain Treacle-negative populations. Intact HeLa cells (*control*) were fixed, stained with IgG antibodies, passed through all FACS-related procedure in the light scattering analysis mode, and used as a control (Treacle-positive). The sorted cell fractions were used for RNA extraction. Reverse transcription-qPCR (RT-qPCR) was performed; it shows levels of 47S pre-rRNA normalized to GAPDH mRNA. Normalized pre-rRNA level in Treacle-positive cells is set to 1. Values are mean ± SD. The calculation is presented for five biological replicates (**, p<0.01 by unpaired t test). (**F**) HeLa cells were processed as described in (**E**). It shows levels of A' site contained unprocessed rRNA normalized to GAPDH mRNA. Normalized unprocessed rRNA level in Treacle-positive cells is set to 1. Values are mean ± SD. The calculation is presented for five biological replicates (**, p<0.01 by unpaired t test).

The online version of this article includes the following figure supplement(s) for figure 1:

**Figure supplement 1.** Treacle depletion induces dispersion and mixing of FC and DFC within nucleoli in both HeLa cells and normal fibroblasts.

**Figure supplement 2.** Depletion of Treacle impairs RNA polymerase I-mediated transcription within the nucleolus.

(*Jaberi Lashkari et al., 2023*). The 11 linkers within the central domain and their adjacent S/E-rich LCRs exhibit high homology. Consequently, the central domain will be referred to as the repeating domain (RD) henceforth. The CD harbors additional NLSs, a UBF binding site, and a lysine-rich LCR (K-rich LCR) at the terminus, which also functions as a nucleolar localization signal (NoLS) (*Figure 2A*).

To evaluate the ability of Treacle to self-assemble into biomolecular condensates, we first investigated its condensation properties using an in vitro system. For this purpose, a fragment of the Treacle protein (amino acids 291–426), encompassing two S/E-rich LCRs and two linker regions, was purified from *Escherichia coli*. In vitro experiments demonstrated that this recombinant fragment formed liquid-like condensates in the presence of 5% polyethylene glycol (*Figure 2B*). These condensates exhibited a spherical morphology and frequently underwent fusion events (*Figure 2C*).

To investigate Treacle's ability to self-organize into biomolecular condensates within cells, we generated fusion proteins by linking Treacle to FusionRed (a non-self-dimerizing fluorescent protein) and *Arabidopsis thaliana* cryptochrome 2 (Cry2), which is known to oligomerize upon exposure to blue light, facilitating the formation of 'optoDroplets' (*Shin and Brangwynne, 2017*). Expression of this opto-Treacle chimeric protein in HeLa cells revealed foci formation in nucleoli in the absence of blue light, consistent with Treacle's natural tendency to occupy FCs (*Figure 2D*; see *Figure 2—figure supplement 1B* for the controls). Under blue light, opto-Treacle formed multiple small foci throughout the nucleoplasm (*Figure 2D*).

Next, we explored Treacle's ability to form biomolecular condensates when overexpressed as a fusion with the far-red fluorescent protein Katushka2S (referred to as 2S; *Shcherbo et al., 2007*) or green fluorescent protein (GFP). At low expression levels (16–24 hr post-transfection), Treacle's fusion protein formed foci only in nucleoli, reflecting its natural occupation of FCs (*Figure 2E*, top panel; *Figure 2—figure supplement 1C*). However, at increased expression levels (48–72 hr post-transfection), it began to form large spherical structures in the nucleoplasm (*Figure 2E*, bottom panel; *Figure 2—figure supplement 1D*). The formation of these structures could not be attributed to the oligomerization of 2S or GFP, as comparable large nuclear condensates were observed at high levels of Treacle expression, even in the absence of a fused fluorescent protein (*Figure 2—figure supplement 1E*). Both intranucleolar and extranucleolar Treacle-2S foci, induced by low and high levels of fusion protein expression respectively, exhibited a spherical shape with an aspect ratio close to one, suggesting susceptibility to surface tension (*Figure 2E*). Transmission electron microscopy confirmed the internal homogeneity of Treacle condensates, showing circular structures with homogeneous protein content without any electron-dense or -light regions inside the condensates (*Figure 2—figure supplement 1F*).

Like the in vitro-formed Treacle condensates, intracellular Treacle condensates demonstrated the ability to fuse. Our observations revealed that nucleolar caps form during AMD-induced (*Figure 2F*;

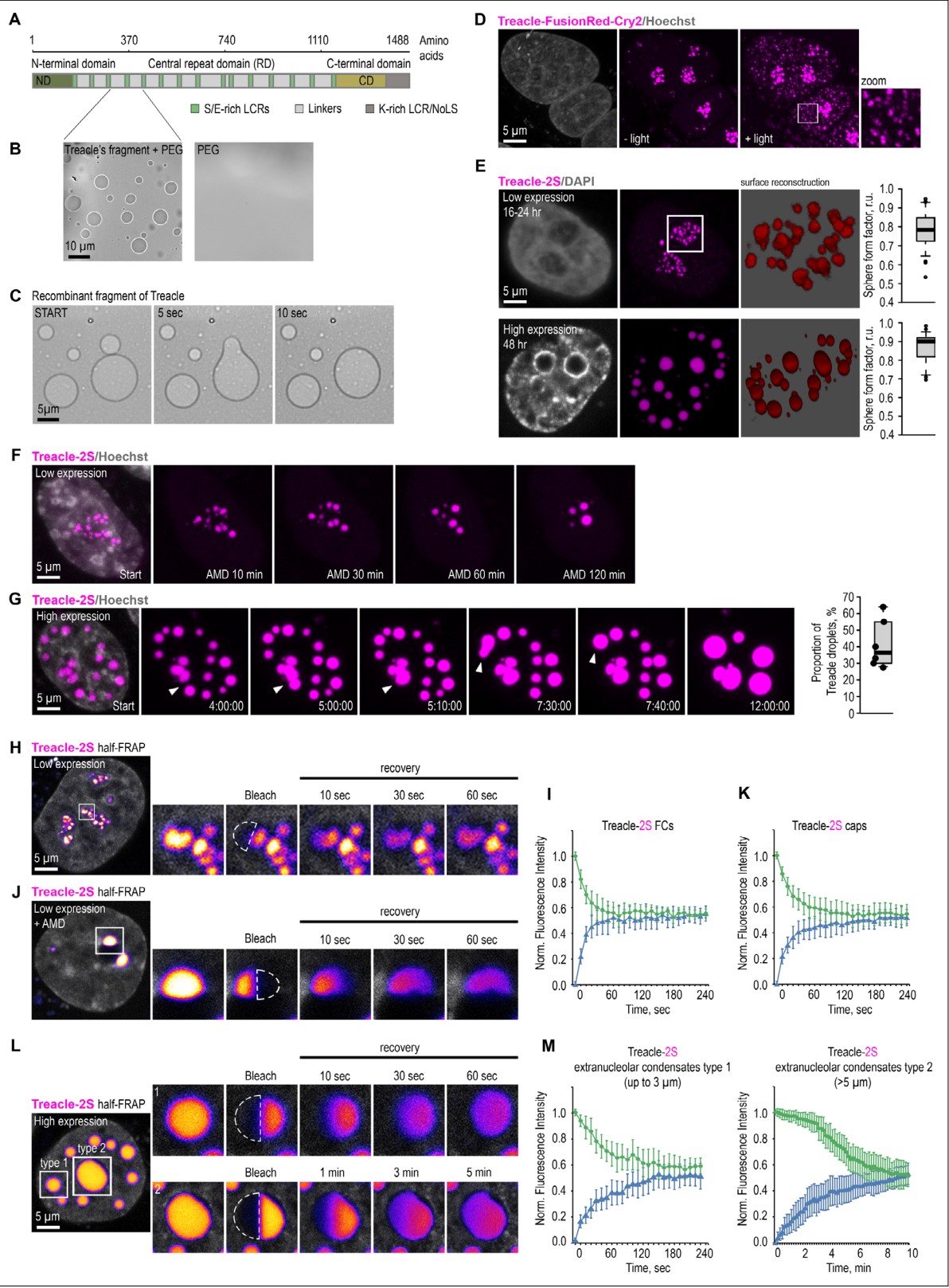

**Figure 2.** Treacle drives the formation of nuclear condensates. (**A**) The structure of the most common isoform of Treacle. Treacle isoform d (1488 amino acids, 152 kDa, NP_001128715.1) is encoded by the Treacle transcript variant 4. It is an intrinsically disordered protein with N-terminal (ND, 1–83 aa), C-terminal (CD, 1121–1488 aa) regions, and central repeated domain (RD, 83–1121 aa) consisting of 15 low-complexity regions (LCRs) interspersed with disordered linker sequences. (**B**) The purified recombinant fragment of Treacle undergoes condensation in vitro in the presence of 5% polyethylene

*Figure 2 continued on next page*

*Figure 2 continued*

glycol. (**C**) Condensates of recombinant Treacle's fragment were time-lapse imaged. Representative images of condensates fusion events are shown. (**D**) HeLa cells were transfected with Treacle-FusionRed-Cry2 (opto-Treacle) construct. DNA was stained with Hoechst 33342 (gray). Formation of Treacle condensates was induced with blue light illumination of the cells for 10 s. Representative images are shown. (**E**) HeLa cells were transfected with Treacle-Katushka2S (Treacle-2S) at a quantity of 50 ng plasmids per $2×10^5$ cells. For low levels of expression analysis, the cells were fixed 16–24 hr after transfection. For high levels of expression analysis, the cells were fixed 48 hr after transfection. DNA was stained with DAPI (gray). Cells were analyzed by laser scanning confocal microscopy. Surface reconstructions of Treacle-2S fibrillar centers (FCs) or extranucleolar condensates are shown in the right panels. Graphs illustrate quantification in arbitrary units of the 3D analysis of Treacle-2S condensate's shape by LasX software. Sphere form factor = sphere area/particle area. (**F**) 16–24 hr after transfection with Treacle-2S, HeLa cells were treated with 0.05 µg/ml actinomycin D (AMD) to induce rDNA transcriptional repression and time-lapse imaged for 120 min. DNA was stained with Hoechst 33342 (gray). (**G**) 48 hr after transfection with Treacle-2S, HeLa cells were live time-lapse imaged for 12 hr. DNA was stained with Hoechst 33342 (gray). Representative images of cells with Treacle-2S condensates fusion events are shown. Graphs illustrate the frequency of fusion Treacle condensates per cell for 12 hr. The calculation is presented for six biological replicates of 10 cells each. (**H**) HeLa cells were transiently transfected with Treacle-2S. 16–24 hr after transfection half-fluorescence recovery after photobleaching (FRAP) analysis of Treacle-labeled fibrillar center (FCs) was performed. Half of the one FC was bleached, and fluorescence recovery was monitored. Representative time-lapse images of the photobleached FC are shown (magnified images). DNA was stained with Hoechst 33342 (gray). (**I**) Graphs illustrate the quantification of the FCs half-FRAP analysis described in (**H**). Half-FRAP curves represent the normalized intensity in the bleached half (blue) and the non-bleached half (green). Each trace represents an average of measurements for at least 20 FCs; error bars represent SD. (**J**) 16–24 hr after transfection with Treacle-2S, HeLa cells were treated with 0.05 µg/ml AMD to induce the formation of nucleolar caps. Half-FRAP analysis of Treacle-labeled nucleolar caps was performed. Representative time-lapse images of photobleached nucleolar caps are shown (magnified images). DNA was stained with Hoechst 33342 (gray). (**K**) Graphs illustrate the quantification of the nucleolar caps half-FRAP analysis described in (**J**). Half-FRAP curves represent the normalized intensity in the bleached half (blue) and the non-bleached half (green). Each trace represents an average of measurements for at least 20 caps; error bars represent SD. (**L**) HeLa cells were transiently transfected with Treacle-2S. 48 hr after transfection, half-FRAP analysis of Treacle-2S extranucleolar condensates was performed. Half of one condensate was bleached, and fluorescence recovery was monitored. The analysis included condensates with diameters of up to 3 µm (type 1), as well as the largest condensates exceeding 5 µm in diameter (type 2). Representative time-lapse images of photobleached Treacle-2S extranucleolar condensates of both categories are shown (magnified images). DNA was stained with Hoechst 33342 (gray). (**M**) Graphs illustrate the quantification of the nucleolar caps half-FRAP analysis described in (**L**). Half-FRAP curves represent the normalized intensity in the bleached half (blue) and the non-bleached half (green). Each trace represents an average of measurements for at least 20 extranucleolar condensates of both categories; error bars represent SD. DNA was stained with Hoechst 33342 (gray).

The online version of this article includes the following video and figure supplement(s) for figure 2:

**Figure 2—video 1.** Intranucleolar Treacle-2S fusion events (low expression, actinomycin D [AMD]-treated cells).
https://elifesciences.org/articles/96722/figures#fig2video1

**Figure 2—video 2.** Extranucleolar Treacle-2S fusion events (high expression).
https://elifesciences.org/articles/96722/figures#fig2video2

**Figure 2—video 3.** Treacle-2S fibrillar centers (FCs) half-fluorescence recovery after photobleaching (FRAP).
https://elifesciences.org/articles/96722/figures#fig2video3

**Figure 2—video 4.** Treacle-2S actinomycin D (AMD) half-fluorescence recovery after photobleaching (FRAP).
https://elifesciences.org/articles/96722/figures#fig2video4

**Figure 2—video 5.** .Treacle-2S intranucleolar condensates type 1 half-fluorescence recovery after photobleaching (FRAP).
https://elifesciences.org/articles/96722/figures#fig2video5

**Figure 2—video 6.** Treacle-2S intranucleolar condensate type 2 half-fluorescence recovery after photobleaching (FRAP).
https://elifesciences.org/articles/96722/figures#fig2video6

**Figure 2—video 7.** Intranucleolar Treacle-2S fusion events (low expression, VP16-treated cells).
https://elifesciences.org/articles/96722/figures#fig2video7

**Figure supplement 1.** Treacle is an intrinsically disordered protein that forms intranuclear condensates upon overexpression.

**Figure supplement 2.** Treacle exhibits limited molecular dynamics between condensates.

*Figure 2—figure supplement 1G*; *Figure 2—video 1*) or DNA damage-induced (*Figure 2—figure supplement 1H*; *Figure 2—video 7*) rDNA transcriptional repression due to the fusion of intranucleolar Treacle-2S foci. Prolonged live-cell imaging further demonstrated that extranucleolar biomolecular condensates from highly overexpressed Treacle-2S frequently fuse, indicating dynamic clustering (*Figure 2G*; *Figure 2—video 2*).

Liquid condensates exhibit a high molecular exchange rate, often assessed through fluorescence recovery after photobleaching (FRAP; *Ganser and Myong, 2020*). To analyze the molecular dynamics of Treacle within its condensates, we employed a combination of full- and half-FRAP methods. Half-FRAP is a specific variation of partial FRAP, where half of the structure of interest is photobleached,

unlike full-FRAP, which involves photobleaching the entire structure (*Muzzopappa et al., 2022*). Half-FRAP analysis revealed an increase in signal intensity in the bleached half of Treacle-2S-formed FCs, accompanied by a proportional decrease in fluorescence in the non-bleached half (*Figure 2H and I*; *Figure 2—video 3*). Full-FRAP analysis of Treacle dynamics within FCs revealed a slow recovery rate (*Figure 2—figure supplement 2A*), suggesting preferential internal mixing. This indicates that molecular exchange occurs between the two halves of the condensate without significant exchange across the boundary separating the condensate from the surrounding phase. Notably, fluorescence in the non-bleached region decreased to approximately half its initial value, suggesting near-complete internal mixing within the coacervate. Comparable full- and half-FRAP dynamics were observed for AMD-induced nucleolar caps *Figure 2J and K*; *Figure 2—figure supplement 2B*; *Figure 2—video 4* and extranucleolar condensates formed under elevated Treacle-2S expression (*Figure 2L and M*; *Figure 2—figure supplement 2C*; *Figure 2—video 5*). Furthermore, the largest extranucleolar condensates exhibited reduced Treacle molecular dynamics, suggesting more gel-like properties (*Figure 2L and M*; *Figure 2—figure supplement 2D*; *Figure 2—video 6*). Presumably, Treacle condensates undergo a liquid-to-gel phase transition over time or upon reaching a critical protein concentration within the condensate. Therefore, using diverse model systems, we illustrated that Treacle can generate biomolecular condensates with key features of structures formed through liquid-like phase separation.

## Treacle's phase separation is regulated by its central domain and CDs

Since we have shown that an RD containing two S/E-rich LCRs can form liquid-like condensates in vitro, we next sought to investigate to what extent this condensation ability of RD is exerted intracellularly and to determine the roles of the CD and ND in intracellular Treacle condensation. For this purpose, we generated Treacle mutants missing the ND ($\Delta$1–83), RD ($\Delta$83–1121), or CD ($\Delta$1121–1488) and overexpressed them in HeLa cells. We then assessed the condensation ability of the overexpressed deletion mutants in wild-type cells using several intracellular condensate models: the formation of (i) FCs, (ii) nucleolar caps under low expression levels (24 hr post-transfection), and (iii) large extranucleolar condensates under high expression levels (48 hr post-transfection).

The $\Delta$1–83 mutant demonstrated condensation properties indistinguishable from full-length Treacle: it concentrated in FCs, formed stable nucleolar caps during transcriptional repression, and, upon increased expression levels, generated numerous large extranucleolar condensates (*Figure 3A*; *Figure 3—figure supplement 1A, D, and E*). Using partial FRAP analysis, we further demonstrated that, similar to full-length Treacle (*Figure 3—figure supplement 1B and C*), the $\Delta$1–83 mutant exhibited a high rate of molecular exchange in all types of model condensates (*Figure 3B and C*).

As expected, the $\Delta$83–1121 mutant exhibited significantly compromised condensation properties. Under low expression levels, it failed to concentrate in FCs (*Figure 3D*; *Figure 3—figure supplement 1F*) and diffusely localized in the nucleolus while still maintaining a high molecular dynamics rate (*Figure 3E and F*). During rDNA transcriptional repression, the $\Delta$83–1121 mutant did not form classical nucleolar caps but partially redistributed between the nucleolar periphery and nucleoplasm, where it began to form condensates (*Figure 3D*; *Figure 3—figure supplement 1F*). Surprisingly, such relocation was associated with a change in its molecular dynamics state. Partial FRAP analysis revealed that nucleoplasmic condensates of the $\Delta$83–1121 mutant, formed due to nucleolar transcriptional repression, began to transition into a solid-like state (*Figure 3E and F*). A similar liquid-solid transition was observed for the $\Delta$83–1121 mutant under high expression levels. In this case, it no longer formed distinct extranucleolar condensates but merged into a unified pan-nuclear solid network (*Figure 3D–F*; *Figure 3—figure supplement 1F*). This behavior suggests that the RD deletion alters Treacle's interaction patterns, making its phase separation properties more reliant on rRNA and other nucleolar proteins.

Unexpectedly, the $\Delta$1121–1488 mutant also failed to condense. When equipped with an artificial NLS from the SV40 virus, it successfully translocated into the nucleus; however, it remained diffusely distributed throughout the nucleoplasm, even under high expression conditions (*Figure 3G*; *Figure 3—figure supplement 1G*). Therefore, without the CD, Treacle cannot undergo autonomous condensation, likely relying on the CD as a nucleation factor. Collectively, these results suggest that the intracellular Treacle condensation depends on a cooperative interaction between RD and CD.

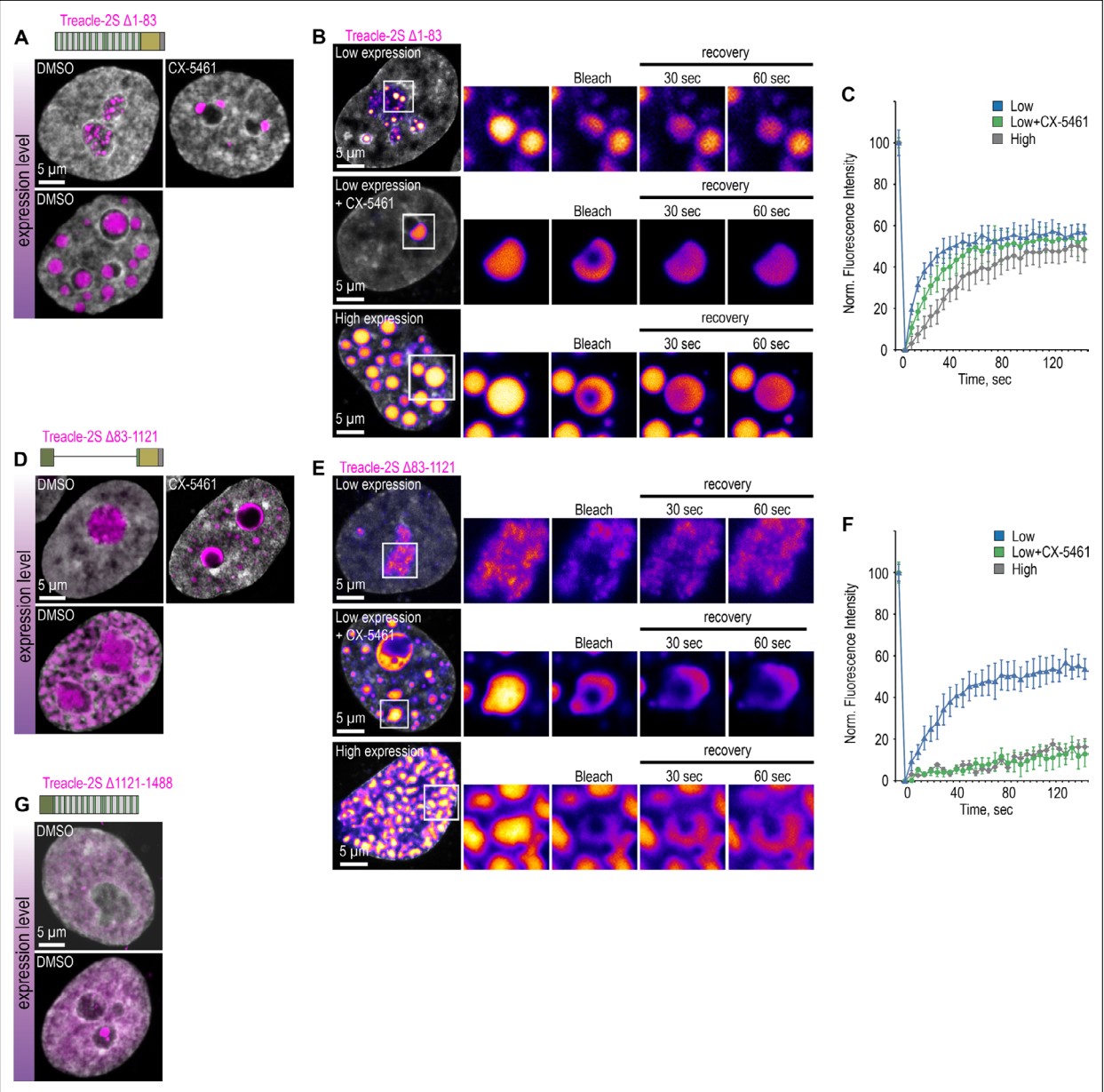

**Figure 3.** Treacle's condensation is regulated by its central and C-terminal domains. (**A**) HeLa cells were transfected with Treacle-2S Δ1–83 deletion mutant. For low or high levels expression analysis cells were cultivated 16–24 or 48 hr after transfection respectively. The expression level is indicated by the colored zone to the left of the cell images. HeLa cells with low expression level were additionally treated with CX-5461 to induce rDNA transcriptional repression and subsequent nucleolar cap formation. Cells were fixed and analyzed by laser scanning confocal microscopy. DNA was stained with DAPI (gray). Representative images of Treacle-2S Δ1–83 condensate are shown. (**B**) HeLa cells were transfected with Treacle-2S Δ1–83 deletion mutant and processed as described in (**A**). Partial fluorescence recovery after photobleaching (FRAP) analysis of Treacle-2S Δ1–83 condensates was performed. A part of each condensate type was photobleached, and the subsequent fluorescence recovery was monitored. Representative time-lapse images of the photobleached condensates are shown (magnified images). DNA was stained with Hoechst 33342 (gray). (**C**) HeLa cells were transfected with Treacle-2S Δ1–83 deletion mutant and processed as described in (**A**). Graphs illustrate the quantification of the Treacle-2S Δ1–83 condensates partial FRAP analysis described in (**B**). Each trace represents an average of measurements for at least 20 Treacle Δ1–83 condensates of each type; error bars represent SD. (**D**) HeLa cells were transfected with Treacle-2S Δ83–1121 deletion mutant and processed as described in (**A**). (**E**) HeLa cells were transfected with Treacle-2S Δ83–1121 deletion mutant and processed as described in (**A**). Partial FRAP analysis of Treacle-2S Δ83–1121 condensates dynamics was performed as described in (**B**). (**F**) HeLa cells were transfected with Treacle-2S Δ83–1121 deletion mutant and processed as described in (**A**). Partial FRAP analysis of 2S-fused Treacle Δ83–1121 condensates dynamics was performed as described in (**B**). Graphs illustrate the quantification of the Treacle-2S Δ83–1121 condensates partial FRAP analysis. Each trace represents an average of measurements for at least 20 Treacle-2S Δ83–1121 condensates of each type; error bars represent SD. (**G**) HeLa cells were transfected with Treacle-2S Δ1121–1488 deletion mutant fused with nuclear localization signal (NLS) from simian virus 40 (SV40). For low or high levels expression analysis cells were cultivated 16–24 or 48 hr

*Figure 3 continued on next page*

*Figure 3 continued*

after transfection respectively. The expression level is indicated by the colored zone to the left of the cell images. Cells were fixed and analyzed by laser scanning confocal microscopy. DNA was stained with DAPI (gray). Representative images of cells are shown.

The online version of this article includes the following figure supplement(s) for figure 3:

**Figure supplement 1.** Treacle's condensation properties are governed by its central repeating domain (RD) and C-terminal domain (CD).

To gain further insights into these phenomena, we investigated the nature of interactions driving Treacle protein condensation. We assessed the behavior of Treacle intracellular condensates after treatment with 1,6-hexanediol (1,6-HD), which disrupts hydrophobic interactions, or high salt concentrations, which interfere with electrostatic interactions. Treatment with the cell-permeable salt ammonium acetate disrupted both FCs and extranucleolar condensates formed at low and high Treacle expression levels, respectively (*Figure 4A*). In contrast, treatment with 1,6-HD did not affect the integrity of Treacle condensates (*Figure 4A*). These observations were further validated using a condensation model based on the recombinant fragment of Treacle's RD in vitro. The efficiency of Treacle's fragment condensation in vitro remained unchanged in the presence of 1,6-HD, whereas increasing salt concentration prevented condensation (*Figure 4B*). Additionally, condensation of the Treacle RD fragment in vitro was sensitive to changes in pH. The optimal efficiency of condensation was observed at pH 7.0, whereas acidification of the medium decreased condensation, and alkalization resulted in the generation of insoluble aggregates (*Figure 4B*). Therefore, it is clear that Treacle condensation, both in vitro and in vivo, is primarily governed by electrostatic interactions rather than hydrophobic forces.

It is reasonable to hypothesize that electrostatic interactions within Treacle condensates are facilitated by the presence of negatively charged S/E-rich LCRs in the RD and positively charged K-rich LCR in the CD. To evaluate this hypothesis, we generated a Treacle mutant lacking the 13S/E-rich LCRs in the RD (ΔSE) and a Treacle mutant missing the 1350–1488 region (ΔNoLS), which includes the K-rich LCR. The deletion of the S/E-rich LCRs significantly compromised Treacle's condensation properties. Like the Δ83–1121 mutant, Treacle ΔSE could not form FCs or nucleolar caps (*Figure 4C*; *Figure 4—figure supplement 1A*), remained in a liquid state within the nucleolus (*Figure 4D and E*), and underwent a liquid-solid phase transition during nucleolar transcriptional repression or high overexpression levels (*Figure 4D and E*). These findings indicate that the RD determines Treacle's condensation properties through the S/E-rich LCRs. In turn, ΔNoLS mutant did not localize to the nucleolus but could form individual perinucleolar condensates (*Figure 4F*; *Figure 4—figure supplement 1B*). Interestingly, the volume of these condensates increased significantly with the expression level of ΔNoLS, but their number did not (*Figure 4F*; *Figure 4—figure supplement 1B*). This observation suggests that, in contrast to full-length Treacle, which forms a substantial number of extranucleolar condensates at high expression levels, ΔNoLS condensates exhibit a limited number of nucleation sites. This implies that the positively charged lysines in the K-rich LCR of the CD act as nucleation points for autonomous Treacle condensate formation. It is also noteworthy that all types of ΔNoLS condensates displayed high molecular dynamics (*Figure 4G and H*). In contrast to full-length Treacle condensates, molecular exchange within ΔNoLS condensates occurred across the boundary separating the coacervate from the surrounding phase, rather than through internal mixing (*Figure 4—figure supplement 1C and D*). This finding reinforces the role of the K-rich LCR as both a nucleation element in Treacle condensate formation and a regulator of Treacle's volumetric dynamics within these condensates.

The disruption of Treacle's condensation properties following the deletion of the S/E-rich LCRs underscores the significance of the negative charge within the central domain for phase separation. However, as noted earlier, the S/E-rich LCRs in the RD are not clustered but rather dispersed throughout the RD, interspersed with linker regions. We propose that for Treacle to exhibit proper condensation behavior, it is not merely the presence of a negative charge in the RD that is critical, both the presence of a negative charge in the RD and its specific spatial distribution within the molecule are critical. Indeed, an analysis of the charge distribution along the Treacle amino acid sequence demonstrated that each S/E-rich LCR, in conjunction with its adjacent linker, within the RD constitutes a strong diblock ampholyte, where a positively charged block (linker) is followed by a negatively charged block (S/E-rich LCR; *Figure 4I*, left panel). In order to determine the physiological importance of this charge distribution, a variant of Treacle (Treacle CS) was created with the same overall net charge but scrambled blocks (*Figure 4I*, right panel). In Treacle CS, regions of opposite

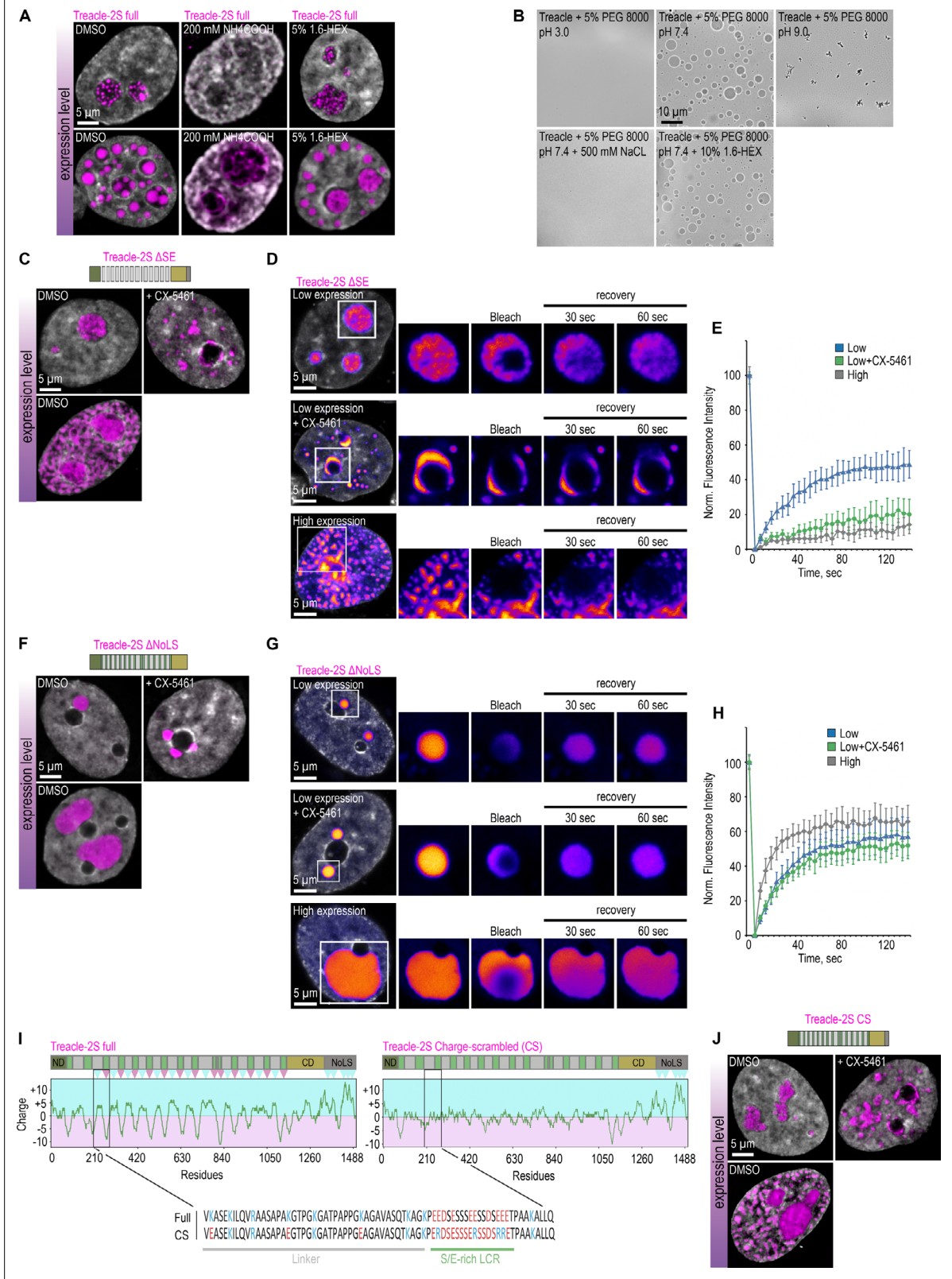

**Figure 4.** The condensation of Treacle is controlled by the specific charge distribution in its domains. (**A**) HeLa cells were transfected with Treacle-2S. For low or high levels expression analysis, cells were cultivated 16–24 or 48 hr after transfection respectively. The expression level is indicated by the colored zone to the left of the cell images. Cells were treated with 5% 1,6-hexandiol (1,6-HEX) for 10 min or 200 mM ammonium acetate for 5 min. Cells were fixed and analyzed by laser scanning confocal microscopy. DNA was stained with DAPI (gray). Representative images of cells are shown. (**B**) The

*Figure 4 continued on next page*

*Figure 4 continued*

purified recombinant fragment of Treacle undergoes condensation in vitro in the presence of 10% 1,6-HEX, 500 mM sodium chloride or buffers with different pH. In each case, 5% PEG 8000 was used as a crowding agent. (**C**) HeLa cells were transfected with Treacle-2S ΔSE mutant. For low or high levels expression analysis cells were cultivated 16–24 or 48 hr after transfection respectively. The expression level is indicated by the colored zone to the left of the cell images. HeLa cells with low expression level were additionally treated with CX-5461 to induce rDNA transcriptional repression. Cells were fixed and analyzed by laser scanning confocal microscopy. DNA was stained with DAPI (gray). Representative images of Treacle ΔSE condensate are shown. (**D**) HeLa cells were transfected with Treacle-2S ΔSE mutant and processed as described in (**C**). Partial fluorescence recovery after photobleaching (FRAP) analysis of Treacle-2S ΔSE condensates was performed. A part of each condensate type was photobleached, and the subsequent fluorescence recovery was monitored. Representative time-lapse images of the photobleached condensates are shown (magnified images). DNA was stained with Hoechst 33342 (gray). (**E**) HeLa cells were transfected with Treacle-2S ΔSE mutant and processed as described in (**C**). Graphs illustrate the quantification of the Treacle-2S ΔSE condensates partial FRAP analysis described in (**D**). Each trace represents an average of measurements for at least 20 Treacle-2S ΔSE condensates of each type; error bars represent SD. (**F**) HeLa cells were transfected with Treacle-2S Δ1350–1488 deletion mutant (Treacle-2S ΔNoLS) and processed as described in (**C**). (**G**) HeLa cells were transfected with Treacle-2S ΔNoLS deletion mutant and processed as described in (**C**). Partial FRAP analysis of Treacle-2S ΔNoLS condensates was performed as described in (**D**). (**H**) HeLa cells were transfected with Treacle-2S ΔNoLS deletion mutant and processed as described in (**C**). Partial FRAP analysis of Treacle-2S ΔNoLS condensates dynamics was performed as described in (**D**). Graphs illustrate the quantification of the Treacle-2S ΔNoLS condensates partial FRAP analysis. Each trace represents an average of measurements for at least 20 Treacle ΔNoLS condensates of each type; error bars represent SD. (**I**) Charge plots of full-length Treacle (left panel) and charge-scrambled Treacle (Treacle CS) form (right panel) are shown. Positive and negative charge blocks are depicted by blue and red triangles, respectively. Charge distribution was calculated as the sum of the charges (Arg and Lys, +1; Glu and Asp, −1;) in the 25 amino acids window range. The center of the panel shows the aligned amino acid sequences of one of the S/E-rich low-complexity region (LCR) and its adjacent linker of RD for both full Treacle and Treacle CS. (**J**) HeLa cells were transfected with Treacle-2S CS mutant and processed as described in (**C**).

The online version of this article includes the following figure supplement(s) for figure 4:

**Figure supplement 1.** The condensation properties of the Treacle are determined by the charge distribution in its central repeating domain (RD).

charge were removed while maintaining the same overall isoelectric point, amino acid composition, and positions of all other residues. As expected, scrambling the charges in the RD fully reproduced the condensation-defective phenotype observed in the Δ83–1121 and ΔSE mutants. Treacle CS failed to form FCs or nucleolar caps, instead assembling into a single pan-nuclear network at high overexpression levels (*Figure 4J*; *Figure 4—figure supplement 1E*). These findings suggest that Treacle's condensation properties depend not only on the net negative charge within the RD but are critically influenced by the specific distribution of charges within this region.

In summary, our findings demonstrate that the cooperative interaction between the RD and CD governs the phase separation properties of Treacle, whereas the ND plays a negligible role in this process. The positively charged K-rich LCR within the CD facilitates its role as a nucleation site for Treacle condensate formation, while the alternating charge blocks in the RD stabilize Treacle condensates, ensuring that Treacle behavior remains independent of subnuclear localization or nucleolar transcriptional activity.

## Treacle condensation is essential for its proper interaction with nucleolar subcompartments

We next aimed to investigate the role of Treacle's condensation properties in its functional interactions with the components of the FC, DFC, and GC within the nucleolus. First, we significantly downregulated endogenous Treacle expression (Treacle kd) using RNA interference (*Figure 5—figure supplement 1A*) and then analyzed the structure organization of the FC, DFC, and GC within the nucleolus in the context of small interfering RNA (siRNA)-resistant full-length or condensation-defective (Δ83–1121 or CS) Treacle mutants.

The expressed full-length Treacle exhibited normal nucleolar localization, similar to endogenous Treacle. It co-localized within the FC with RPA194 and UBF, while FBL was localized in the DFC at the periphery of the FC, and NCL and B23 surrounded the DFC (*Figure 5A*). In contrast, expression of the Δ83–1121 or CS Treacle mutants led to the dispersal and mixing of FC and DFC components within the nucleolus (*Figure 5B*; *Figure 5—figure supplement 1B*). Despite this, the Δ83–1121 or CS mutants still co-localized with UBF1, but to a significantly lesser extent with RPA194, suggesting a potential loss of their binding interaction. Interestingly, these mutants showed partial overlap with NCL, but not with B23 (*Figure 5B*, *Figure 5—figure supplement 1B*). These observations suggest that disrupting Treacle's condensation properties may alter its interaction preferences.

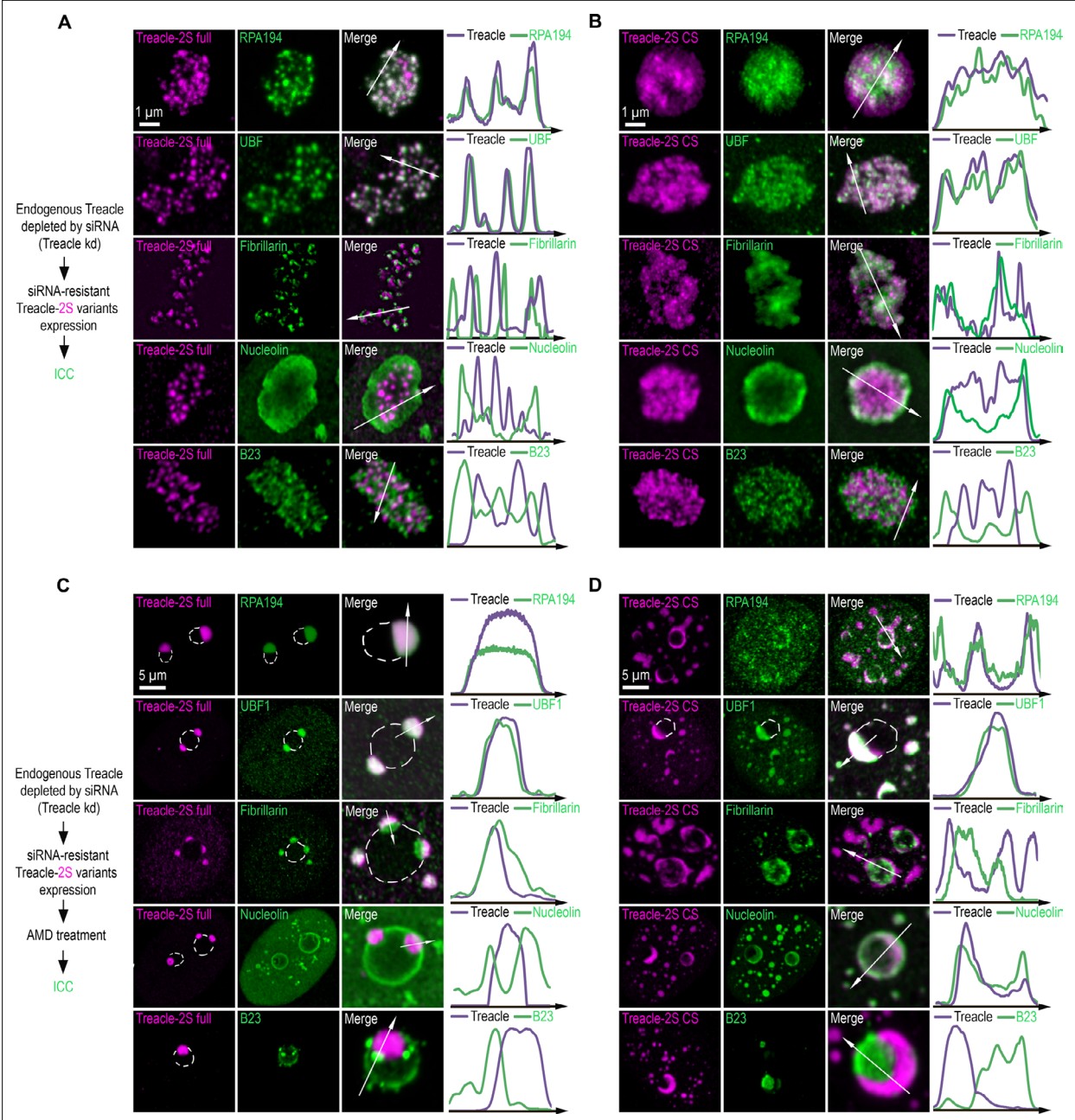

**Figure 5.** The condensation of Treacle facilitates its functional interactions with partner proteins. (**A**) Endogenous Treacle was depleted by small interfering RNA (siRNA)-mediated knockdown (Treacle kd). Next, Treacle-depleted cells were transfected with siRNA-resistant full-length Treacle-2S (Treacle-2S full). Cells were fixed 16–24 after transfection and immunostained with either RPA194, UBF, fibrillarin, B23, or nucleolin antibodies. Cells were analyzed by laser scanning confocal microscopy. Representative images of magnified nucleoli are shown. Co-localization analysis was performed on the merged images. Graphs illustrate quantification in arbitrary units of Treacle-2S variants and RPA194, UBF, fibrillarin, B23, or nucleolin fluorescence distribution along the lines shown in the figures. (**B**) Endogenous Treacle was depleted by siRNA-mediated knockdown (Treacle kd). Next, Treacle-depleted cells were transfected with siRNA-resistant charge-scrambled Treacle-2S (Treacle-2S CS) and processed as described in (**A**). (**C**) Endogenous Treacle was depleted by siRNA-mediated knockdown (Treacle kd). Next, Treacle-depleted cells were transfected with siRNA-resistant full-length Treacle-2S (Treacle-2S full). 16–24 hr after transfection cells were treated with 0.05 μg/ml actinomycin D (AMD) to induce rDNA transcriptional repression, fixed and processed as described in (**A**). (**D**) Endogenous Treacle was depleted by siRNA-mediated knockdown (Treacle kd). Next, Treacle-depleted cells were transfected with siRNA-resistant charge-scrambled Treacle-2S (Treacle-2S CS). 16–24 hr after transfection cells were treated with 0.05 μg/ml AMD to induce rDNA transcriptional repression, fixed and processed as described in (**A**).

The online version of this article includes the following source data and figure supplement(s) for figure 5:

**Figure supplement 1.** Disruption of Treacle's condensation properties alters its interaction specificity with protein partners.

*Figure 5 continued on next page*

*Figure 5 continued*

**Figure supplement 1—source data 1.** TIFF file containing original western blots for panel A, indicating the relevant bands.

**Figure supplement 1—source data 2.** Original files for western blot analysis displayed in panel A.

This hypothesis is further supported by the analysis of the localization of nucleolar components and condensation-defective Treacle forms following AMD-induced nucleolar transcriptional repression. After AMD treatment, full-length Treacle co-localized with UBF1 and RPA194 at the nucleolar caps (*Figure 5C*). FBL partially merged with the caps, while NCL and B23 were completely displaced to the nucleolar rim and nucleoplasm (*Figure 5C*). In contrast, the AMD-induced perinucleolar or nucleoplasmic solid condensates formed by the Δ83–1121 or CS Treacle mutants still co-localized with UBF1, but to a significantly lesser extent with RPA194 (*Figure 5D*; *Figure 5—figure supplement 1C*). Interestingly, these condensates exhibited behavior opposite to that of full-length Treacle condensates with respect to FBL and NCL: they displaced FBL and incorporated NCL (*Figure 5D*; *Figure 5—figure supplement 1C*). The patterns of association between FC/DFC/GC proteins and the extranucleolar condensates of Δ83–1121 and CS Treacle mutants at high expression levels exhibited similar alterations when compared to those observed with full-length Treacle (*Figure 5—figure supplement 1D and E*).

Thus, the disruption of Treacle's condensation properties, either through RD deletion or charge scrambling, results in a shift in its interaction repertoire. Specifically, while interaction with UBF1 is preserved, interactions with RNA Pol I and FBL are notably diminished, whereas a new interaction with NCL is established. It is likely that the correct condensation of Treacle contributes to the formation of the cooperative FC/DFC structure by mediating functional interactions with partner proteins.

## The condensation of Treacle regulates rRNA transcription and processing

In the above sections, we demonstrated that Treacle supports both rRNA transcription and processing. We hypothesized that this multifunctionality of Treacle could be related to its ability to condense and the formation of the cooperative FC/DFC structure. To test this hypothesis, we significantly downregulated endogenous Treacle expression (Treacle kd) using RNA interference and then analyzed rRNA transcription levels in the context of siRNA-resistant full-length or condensation-defective (Δ83–1121 or CS) Treacle mutants. We found that the expression of condensation-defective Treacle mutants significantly reduced the transcription of rRNA compared to full-length Treacle (*Figure 6A*). This effect correlated with a reduced ability of condensation-defective Treacle mutants to maintain the association of partner proteins RPA194 and UBF1 at the rDNA promoter (*Figure 6C*), along with diminished intrinsic binding of these mutants to rDNA (*Figure 6B*).

These observations suggest that the condensation properties of Treacle underlie its ability to concentrate components of the transcriptional machinery in the FC and separate them from the DFC. Moreover, such separation is likely required to maintain highly efficient transcription of rDNA. We hypothesized that the same principle could regulate the role of Treacle in rRNA processing. To test this assumption, we depleted the level of endogenous Treacle using RNA interference and then analyzed the spatial localization of newly synthesized 47S pre-rRNA and its processing level in cells expressing siRNA-resistant full-length or condensation-defective (Δ83–1121 or CS) Treacle mutants. Microscopy analyses with fluorescence in situ hybridization (FISH)-labeled rRNA revealed that newly synthesized 47S pre-rRNA transcripts clustered at the periphery of FCs in full-length Treacle-expressing cells, reflecting the radial flow of rRNA from the FC to the GC (*Figure 6D*). Newly synthesized rRNA was also relocated to the periphery of large Treacle condensates formed at high expression levels (*Figure 6E*). However, no apparent clustering of newly synthesized rRNA was observed in cells expressing condensation-defective Treacle mutants. Instead, the newly synthesized transcripts were diffusely mixed with Δ83–1121 and CS Treacle (*Figure 6D*). Nascent rRNA transcripts were similarly mixed with delocalized UBF in cells depleted of endogenous Treacle (Treacle kn; *Figure 6F*). Finally, reverse transcription-quantitative polymerase chain reaction (RT-qPCR) analysis confirmed that cells overexpressing condensation-defective Treacle mutants had impaired 5' external transcribed spacer processing compared to cells overexpressing full-length Treacle (*Figure 6G*). Therefore, it can be concluded that the mixing of FC and DFC components due to the expression of condensation-defective

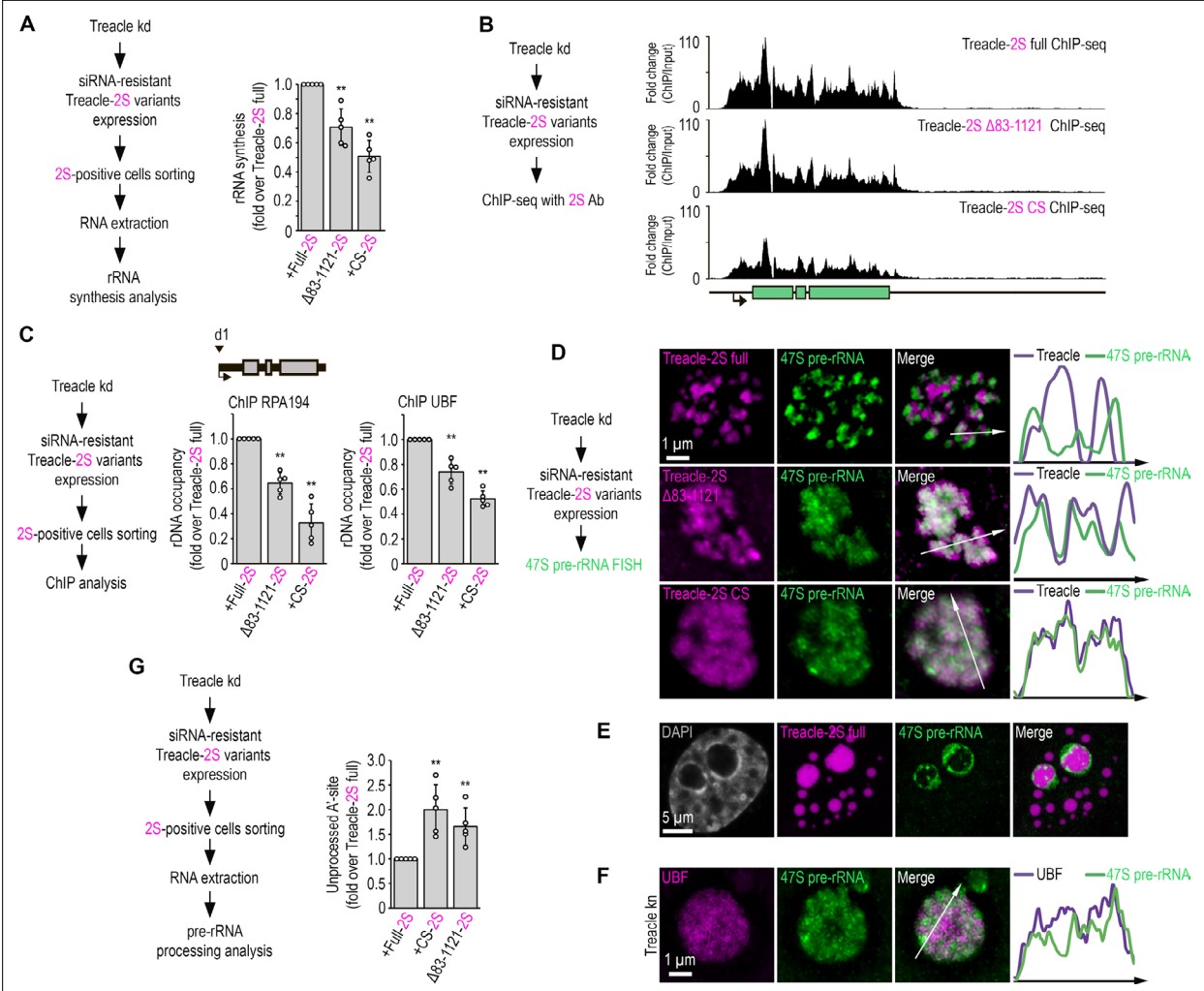

**Figure 6.** The condensation of Treacle is essential for the transcription and processing of ribosomal RNA (rRNA). (**A**) Endogenous Treacle was depleted by small interfering RNA (siRNA)-mediated knockdown (Treacle kd). Next, Treacle-depleted cells were transfected with siRNA-resistant full-length Treacle-2S (Treacle-2S full), charge-scrambled Treacle-2S (Treacle-2S CS), or Treacle-2S Δ83–1121 deletion mutant (Treacle-2S Δ83–1121). Cells were fixed 16–24 after transfection and subjected to cell sorting in the fluorescent analysis mode to obtain 2S-positive populations. The sorted cell fractions were used for RNA extraction. Reverse transcription-quantitative polymerase chain reaction (RT-qPCR) was performed; it shows levels of 47S pre-rRNA normalized to GAPDH mRNA. Normalized pre-rRNA level in full-length Treacle-2S-positive cells is set to 1. Values are mean ± SD. The calculation is presented for five biological replicates. **, p<0.01 by unpaired t test. (**B**) HeLa cells were processed as described in (**A**). Cells were fixed 16–24 after transfection and subjected to chromatin immunoprecipitation sequencing (ChIP-seq) analysis with Katushka2S antibodies. ChIP-seq signal were normalized to the input. (**C**) HeLa cells were processed as described in (**A**). Cells were fixed 16–24 after transfection and subjected to cell sorting in the fluorescent analysis mode to obtain 2S-positive populations. The sorted cell fractions were used for used for ChIP with RPA194 or UBF antibodies. ChIP was followed by qPCR using the d1 primers to the promoter of the rRNA gene (positioned as indicated on the scheme). Percentage of enrichment relative to input in full-length Treacle-2S-positive cells is set to 1. Values are mean ± SD. The calculation is presented for five biological replicates. **, p<0.01 by unpaired t test. (**D**) HeLa cells were processed as described in (**A**). Cells were fixed 16–24 after transfection and stained for 47S pre-rRNA (revealed by single-molecule fluorescence in situ hybridization [smFISH]). Cells were analyzed by laser scanning confocal microscopy. Representative images of magnified nucleoli are shown. Co-localization analysis was performed on the merged images. Graphs illustrate quantification in arbitrary units of Treacle-2S variants and smFISH fluorescence distribution along the lines shown in the figures. (**E**) HeLa cells were transfected with full-length Treacle-2S. 48 hr after transfection cells were fixed and stained for 47S pre-rRNA (revealed by smFISH). Cells were analyzed by laser scanning confocal microscopy. Representative images of magnified nucleoli are shown. (**F**) HeLa cells were transfected with a construct coding CRISPR/Cas9 and single guide RNA (sgRNA) to the TCOF1 gene (Treacle kn). 7–10 days after transfection, the cells were fixed and stained with UBF antibodies and 47S pre-rRNA. Representative images of Treacle-negative nucleolus (magnified images) are shown. (**G**) HeLa cells were processed as described in (**A**). Cells were fixed 16–24 after transfection and subjected to cell sorting in the fluorescent analysis mode to obtain 2S-positive populations. The sorted cell fractions were used for RNA extraction. RT-qPCR was performed; it shows levels of A' site contained unprocessed rRNA normalized to GAPDH mRNA. Normalized unprocessed rRNA level in full-length Treacle-2S-positive cells is set to 1. Values are mean ± SD. The calculation is presented for five biological replicates. **, p<0.01 by unpaired t test.

Treacle or the depletion of endogenous Treacle disrupts the directional traffic of nascent rRNA from the FC to the DFC and further to the GC, potentially causing inefficient processing.

Our results indicate that Treacle's condensation properties separate the FC and DFC, leading to the spatial segregation of rRNA synthesis and subsequent processing. The mixing of FC and DFC components due to the disruption of Treacle's condensation ability reduces the efficiency of both processes, with equivalent effects to the complete depletion of endogenous Treacle.

## The condensation of Treacle is essential for DDR activation in ribosomal genes under genotoxic stress

Previous studies have demonstrated that Treacle is critical for inducing the DDR in ribosomal genes under certain types of stress (*Korsholm et al., 2019*; *Larsen et al., 2014*; *Mooser et al., 2020*; *Velichko et al., 2019*). Here, we aimed to investigate the contribution of Treacle's condensation to rDNA damage response. We induced rDNA damage using the widely used chemotherapeutic drug etoposide (VP16), which acts as a topoisomerase II inhibitor, inducing double-strand breaks (*Bax et al., 2019*). Treating cells with 90 µM VP16 for 30 min resulted in the rapid recruitment of TOPBP1 to nucleoli and its co-localization with Treacle in HeLa (*Figure 7A*; *Figure 7—figure supplement 1A*) and MCF7 cells (*Figure 7—figure supplement 1B*). Furthermore, proximity ligation assay (PLA) analysis conducted after DNA damage induction provided evidence of a physical association between TOPBP1 and Treacle (*Figure 7B*; *Figure 7—figure supplement 1C*). Reducing endogenous Treacle levels, either through RNA interference (Treacle kd) or CRISPR/Cas9-mediated depletion (Treacle kn), effectively blocked the relocalization of TOPBP1 to nucleoli (*Figure 7D*; *Figure 7—figure supplement 2A and B*) and its occupancy of rDNA (*Figure 7C*). Interestingly, with Treacle kd, a substantial amount of Treacle still remained in nucleoli (*Figure 7D*). However, this amount was insufficient to facilitate its interaction with TOPBP1, likely due to the need for specific stoichiometric ratios for efficient interaction.

Next, we investigated whether the interaction between Treacle and TOPBP1 in Treacle kd cells could be restored by overexpressing different Treacle variants. Immunocytochemical analysis and ChIP with antibodies against TOPBP1 indicated that expressing (siRNA)-resistant full-length Treacle in Treacle kd cells effectively restored the VP16-induced interaction between Treacle and TOPBP1 (*Figure 7E*; *Figure 7—figure supplement 3A*), as well as the enrichment of TOPBP1 at the promoters of rDNA (*Figure 7F*). However, expressing (siRNA)-resistant condensation-defective forms of Treacle (Δ83–1121 or CS) did not lead to such restoration (*Figure 7E and F*; *Figure 7—figure supplement 3A*).

The response to DNA damage involves the activation of various signaling kinases and the recruitment of different repair factors to the DNA break site. ChIP followed by qPCR with antibodies against a panel of DNA repair proteins confirmed that VP16 induced DDR at the promoters of rDNA, including the activation of ATM serine/threonine kinase (ATM) and ATR, phosphorylation of the H2A.X variant histone (H2AX), and recruitment of repair factors such as tumor protein p53 binding protein 1 (TP53BP1/53BP1) and BRCA1 DNA repair associated (BRCA1; *Figure 7G*; *Figure 7—figure supplement 3C*). As expected, nucleolar DDR was fully controlled by the interaction between Treacle and TOPBP1, as the knockdown of either of these proteins dramatically reduced VP16-induced repair signaling at rDNA (*Figure 7G*; *Figure 7—figure supplement 3B and C*). Finally, it was confirmed that nucleolar DDR could be efficiently restored in Treacle kd cells by overexpressing (siRNA)-resistant full-length Treacle but not its condensation-defective mutants (Δ83–1121 or CS; *Figure 7H*; *Figure 7—figure supplement 3D*). Therefore, we can conclude that Treacle's phase separation is crucial in facilitating its binding to TOPBP1 and activating the DDR in response to genotoxic damage to rDNA.

## Discussion

The nucleolus is a multicomponent phase condensate formed on the platform of rRNA, which is processed and incorporated into assembling ribosomes simultaneously with migration from FCs to DFCs and further to GCs. The constitutive components of DFCs and GCs are NCL and NPM1, respectively (*Feric et al., 2016*; *Lafontaine et al., 2021*). Which protein plays a comparable role in FCs continues to be debated. Our results suggest that the phosphoprotein Treacle serves as the scaffold protein for nucleolar FCs in human cells. This conclusion aligns entirely with the observation that

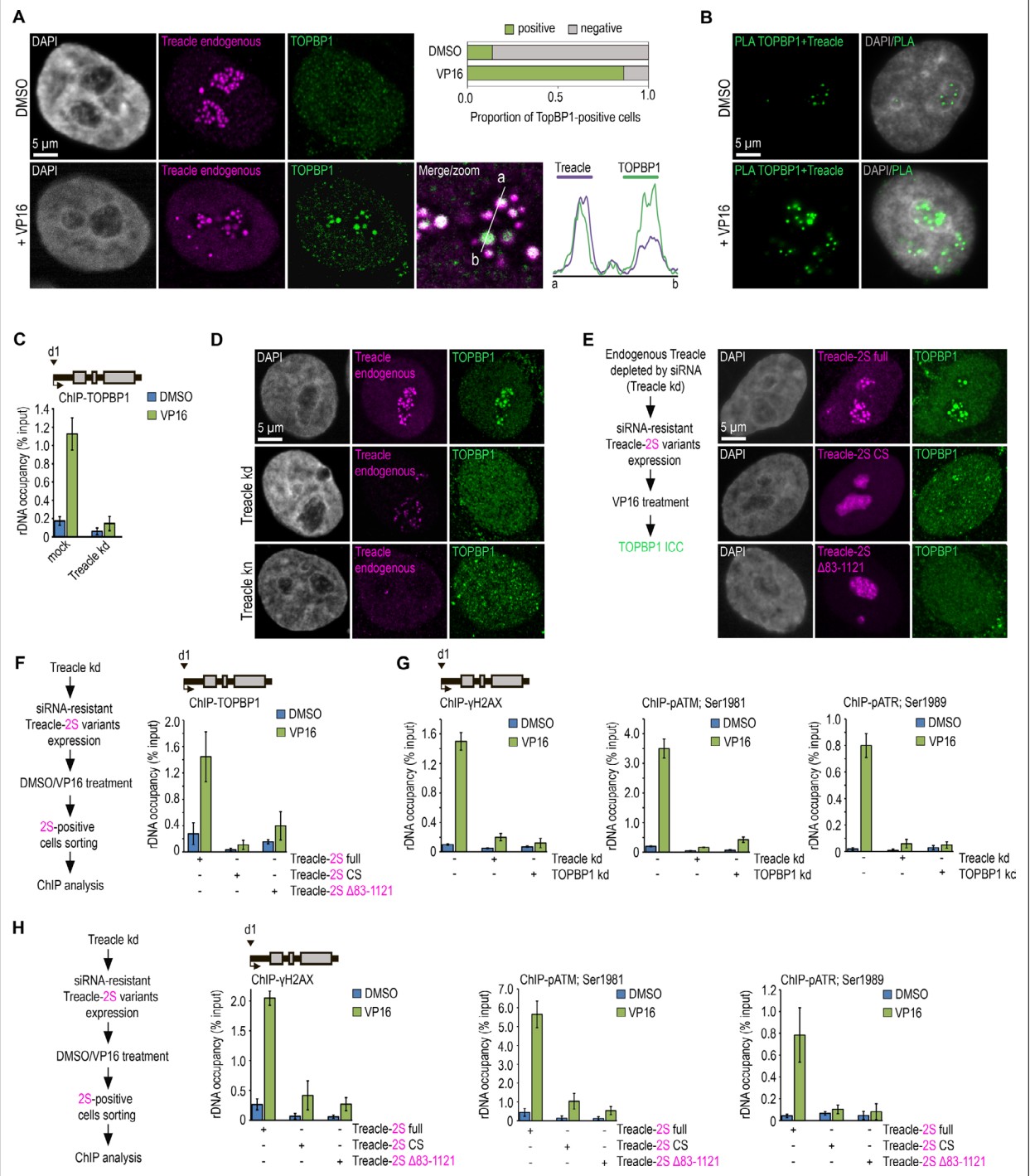

**Figure 7.** Treacle phase separation is essential for DNA damage response (DDR) activation in ribosomal genes under genotoxic stress conditions. (**A**) DMSO-treated and VP16-treated (90 μM, 30 min) HeLa cells were co-immunostained for Treacle (Treacle endogenous; magenta) and TOPBP1 (green) and analyzed by laser scanning confocal microscopy. The DNA was stained with DAPI (gray). Co-localization analysis was performed on the merged images (magnified images). Graphs illustrate quantification in arbitrary units of Treacle and TOPBP1 fluorescence distribution along the lines shown in the figures. Percentage of cells containing TOPBP1 (TOPBP1-positive) foci within nucleoli is shown. (**B**) DMSO-treated and VP16-treated (90 μM, 30 min) HeLa cells were subjected to proximity ligation assay (PLA) with antibodies against TOPBP1 and Treacle. DNA was stained with DAPI (gray). PLA detection of Treacle-TOPBP1 interactions is visible as distinct green fluorescent dots. (**C**) Mock-treated HeLa cells or cells with small interfering RNA (siRNA)-mediated Treacle knockdown (Treacle kd) were treated with DMSO or VP16 (90 μM, 30 min). Chromatin immunoprecipitation (ChIP) experiments were performed with antibodies against TOPBP1. ChIP was followed by quantitative polymerase chain reaction (qPCR) using the d1 primers to the promoter of the ribosomal RNA (rRNA) gene (positioned as indicated on the scheme). Data are represented relative to the input. Values are means ±

*Figure 7 continued on next page*

*Figure 7 continued*

SD from at least three independent replicates. (**D**) Intact HeLa cells, siRNA-depleted for Treacle (Treacle kd) cells, or CRISPR/Cas9-depleted for Treacle (Treacle kn) cells were treated with 90 µM VP16 for 30 min. Cells were co-immunostained for TOPBP1 (green) and Treacle (magenta) antibodies and analyzed by laser scanning confocal microscopy. The DNA was stained with DAPI (gray). (**E**) Endogenous Treacle was depleted by siRNA-mediated knockdown (Treacle kd). Next, Treacle-depleted cells were transfected with plasmid constructs encoding either siRNA-resistant full-length Treacle-2S (Treacle-2S full), Treacle-2S Δ83–1121 deletion mutant, or charge-scrambled mutant Treacle-2S (Treacle-2S CS). 24 hr after transfection, cells were treated with VP16 (90 µM for 30 min), fixed and stained for TOPBP1 (green), and analyzed by laser scanning confocal microscopy. The DNA was stained with DAPI (gray). (**F**) Endogenous Treacle was depleted by siRNA-mediated knockdown (Treacle kd). Treacle-depleted cells were transfected with plasmid constructs encoding either siRNA-resistant Treacle-2S full, Treacle-2S Δ83–1121, or Treacle-2S CS. 24 hr after transfection, cells were treated with DMSO or VP16 (90 µM for 30 min) and fixed. Cells were subjected to cell sorting in the fluorescent analysis mode to obtain 2S-positive populations. At least $2 \times 10^6$ sorted cells were used for ChIP with TOPBP1 antibodies. ChIP was followed by qPCR using the d1 primers to the promoter of the rRNA gene (positioned as indicated on the scheme). Data are represented relative to the input. Values are means ± SD from at least three independent replicates. (**G**) Intact HeLa cells and cells siRNA-depleted for either Treacle (Treacle kd) or TOPBP1 (TOPBP1 kd) were treated with DMSO or 90 µM VP16 for 30 min. ChIP experiments were performed with antibodies against phospho-ATR (pATR; Ser1989), phospho-ATM (pATM; Ser1981), or γH2AX antibodies. ChIP was followed as described in (**F**). (**H**) HeLa cells were processed as described in (**F**). At least $2 \times 10^6$ sorted cells were used for ChIP with phospho-ATR (pATR; Ser1989), phospho-ATM (pATM; Ser1981), or γH2AX antibodies. ChIP was followed by qPCR using the d1 primers to the promoter of the rRNA gene (positioned as indicated on the scheme). Data are represented relative to the input. Values are means ± SD from at least three independent replicates.

The online version of this article includes the following source data and figure supplement(s) for figure 7:

**Figure supplement 1.** Etoposide (VP16) induces the association of TOPBP1 with Treacle in HeLa and MCF7 cells.

**Figure supplement 2.** Etoposide (VP16) promotes TOPBP1 translocation to the nucleolus in a Treacle-dependent manner.

**Figure supplement 3.** Disruption of Treacle's condensation properties abrogates its interaction with TOPBP1 and impairs activation of the rDNA damage response under genotoxic stress.

**Figure supplement 3—source data 1.** TIFF file containing original western blots for panel B, indicating the relevant bands.

**Figure supplement 3—source data 2.** Original files for western blot analysis displayed in panel B.

the evolutionary emergence of Treacle correlates with the appearance of the three-partite nucleolus in amniotes (*Jaberi Lashkari et al., 2023*). Introducing human Treacle into zebrafish, which lack a competent ortholog for FC formation, induced the reorganization of the nucleolar structure from bipartite to tripartite, accompanied by the formation of FC-like structures.

Our findings demonstrate that the condensation properties of Treacle are governed by the synergistic interplay between its RD and CD. While the RD can form liquid-like condensates alone in vitro, its condensation within the cellular environment requires the positively charged CD, which likely functions as a nucleation platform. Notably, the zebrafish ortholog of Treacle exhibits sequence homology with human Treacle exclusively within the RD (*Hill-Terán et al., 2024*), lacking a homologous sequence in the CD. This structural difference likely accounts for its inability to undergo condensation and form FC-like structures.

Treacle functions as a block polyampholyte, with its condensation properties governed by the alternating charges of the CD and the strongly positive charge of the CD. These properties suggest that its condensation is likely driven by coacervation or self-coacervation mechanisms. Complex coacervation refers to a phase transition process mediated by polyelectrolyte complexation, where oppositely charged macromolecules associate through electrostatic interactions and phase-separate from the surrounding solution to form dense, charge-balanced structures known as coacervates (*Choi et al., 2020*; *Sathyavageeswaran et al., 2024*; *Sing and Perry, 2020*; *Pappu et al., 2023*). The driving forces for complex coacervation can stem from homotypic or heterotypic interactions between charged polymers (*Sanders et al., 2020*). Our experimental findings suggest that both mechanisms may be drivers of Treacle's coacervation.

Homotypic coacervation of Treacle may occur through a two-step process: nucleation is facilitated by electrostatic interactions between the lysine (K)-rich LCR in the CD and the glutamic acid (E)-rich LCR in the RD, while fluidization is achieved via dynamic interactions between the E-rich LCR and K-rich linkers, primarily within the RD. Conversely, heterotypic coacervation of Treacle may involve interactions with other polyampholyte proteins enriched in the FCs, many of which are known Treacle partners and possess the ability to facilitate coacervation. For example, it is hypothesized that the CD of Treacle may promote nucleation through interactions with UBF, which contains extended aspartic acid (D)/glutamic acid (E) tracts and demonstrates autonomous condensation properties in the presence

of rDNA (*Gál et al., 2022*; *King et al., 2024*; *Lin and Yeh, 2009*). Additionally, Treacle interacts with RNA Pol I, whose subunits (e.g. POLR1F and POLR1G) are characterized by prominent K-rich regions and E-rich tracts (*King et al., 2024*). These regions are likely to facilitate electrostatic interactions between RNA Pol I and Treacle, potentially modulating its condensation dynamics within the cellular environment. This hypothesis is further supported by our experimental observations showing that charge scrambling or deletion of Treacle's RD partially disrupts its interaction with RNA Pol I subunit RPA194, reduces its recruitment to rDNA, and impairs its condensation properties. Together, these findings highlight the interplay of homotypic and heterotypic interactions in driving Treacle's coacervation and its functional role within the nucleolus.

Endogenous Treacle may undergo post-translational modifications. Specifically, the RD of Treacle is known to be heavily phosphorylated and ubiquitinated (*Mooser et al., 2020*; *Werner et al., 2015*; *Werner et al., 2018*). Currently, the role of these modifications is primarily attributed to facilitating interactions with partner proteins such as the cullin 3 (CUL3), kelch repeat, and BTB domain containing 8 (KBTBD8) complex or nucleolar and coiled-body phosphoprotein 1 (NOLC1; *Werner et al., 2018*). However, it is known that phosphorylation can modulate phase separation, particularly by enhancing or reducing the polyampholytic properties of disordered regions (*Yamazaki et al., 2022*). We propose that post-translational modifications, such as ubiquitination and phosphorylation, fine-tune the condensation properties of Treacle. These modifications may do so by modulating interactions with binding partners and altering the ampholytic properties of the RD through changes in its charge ratio.

The 47S pre-rRNA is synthesized at the FC/DFC boundary, while its primary processing begins in the DFC. DFC formation is mediated by the phase separation of FBL gathered on the rRNA scaffold and is maintained as long as there is a directed radial flow of rRNA from the FC to the DFC (*Wu et al., 2021*; *Yao et al., 2019*). While the role of Treacle in ensuring rRNA processing has been repeatedly described (*Calo et al., 2018*; *Gonzales et al., 2005*), the underlying mechanism remains unclear. We demonstrated that the efficiency of rRNA processing is compromised not only by the complete depletion of endogenous Treacle but also by the expression of condensation-defective Treacle mutants. This impairment correlates with a loss of Treacle association with fibrillarin and, conversely, an increased association of Treacle with newly synthesized rRNA and nucleolin. This shift likely disrupts the cooperative organization of the FC/DFC and the mixing of their components. The resulting disorganization and mixing of FC and DFC components, driven by the loss of Treacle's condensation properties, may compromise the radial flow of rRNA, thereby hindering its normal processing.

We also explored the role of Treacle's phase separation in nucleolar DDR. Treacle's well-known partner, TOPBP1, plays a crucial role in activating ATR during DNA replication stress (*Day et al., 2021*). The interaction between TOPBP1 and Treacle is essential for nucleoli-specific DDR and replication stress response (*Mooser et al., 2020*; *Velichko et al., 2019*). This interaction occurs via phosphoserines 1227 and 1228 in a conserved acidic motif in the CD of Treacle (*Mooser et al., 2020*). However, we showed that Treacle with intact phosphoserines 1227/1228 but unable to undergo condensation loses the ability to interact with TOPBP1 during genotoxic stress. This observation implies that Treacle's interaction with TOPBP1 and the subsequent activation of nucleolar DDR requires Treacle's phase separation ability.

In conclusion, our study revealed Treacle's role not only as a structural scaffold for the FC/DFC but also as a nucleolar hub protein that integrates functions in rDNA transcription, rRNA processing, and rDNA repair.

# Materials and methods

## Key resources table

| Reagent type (species) or resource | Designation | Source or reference | Identifiers | Additional information |
|---|---|---|---|---|
| Cell line (*Homo sapiens*) | HeLa | ATCC | CCL-2, RRID:CVCL_0030 | Mycoplasma free |
| Cell line (*Homo sapiens*) | MCF7 | ATCC | HTB-22, RRID:CVCL_0031 | Mycoplasma free |
| Cell line (*Homo sapiens*) | Dermal fibroblast (normal, Adult) | Gift from Dr. M Lagarkova | | Female 46XX, Mycoplasma free |
| Chemical compound, drug | Etoposide (VP16) | Sigma-Aldrich | E1383 | |

*Continued on next page*

*Continued*

| Reagent type (species) or resource | Designation | Source or reference | Identifiers | Additional information |
|---|---|---|---|---|
| Chemical compound, drug | CX-5461 | Sigma-Aldrich | #5092650001 | |
| Chemical compound, drug | Actinomycin D (AMD) | Sigma-Aldrich | A1410 | |
| Chemical compound, drug | 1,6-Hexanediol (1,6-HD) | Sigma-Aldrich | #240117 | |
| Chemical compound, drug | Ammonium acetate | Serva | #39750.01 | |
| Chemical compound, drug | Hoechst 33342 | Sigma-Aldrich | #14533 | |
| Chemical compound, drug | DAPI | Sigma-Aldrich | D9542 | |
| Antibody | Anti-Treacle/TCOF1 (Mouse monoclonal) | Santa Cruz Biotechnology | sc-374536, RRID:AB_10987865 | ICC (1:500), WB (1: 5000), ChIP-qPCR |
| Antibody | Anti-Treacle/TCOF1 (Rabbit polyclonal) | Sigma-Aldrich | HPA038237 | ICC (1:500) |
| Antibody | Anti-RPA194 (Mouse monoclonal) | Santa Cruz Biotechnology | sc-48385, RRID:AB_675814 | ICC (1:50), ChIP-qPCR |
| Antibody | Anti-UBF1 (Rabbit polyclonal) | Thermo Fisher | PA5-82245, RRID:AB_2789405 | ICC (1:200), ChIP-qPCR |
| Antibody | Anti-Fibrillarin (Rabbit monoclonal) | Abcam | ab166630, RRID:AB_2928100 | ICC (1:500) |
| Antibody | Anti-Nucleolin (Rabbit monoclonal) | Cell Signaling | 14574, RRID:AB_2798519 | ICC (1:500) |
| Antibody | Anti-B23 (Mouse monoclonal) | Sigma-Aldrich | B0556, RRID:AB_2154872 | ICC (1:300) |
| Antibody | Anti-TOPBP1 (Mouse monoclonal) | Santa Cruz Biotechnology | sc-271043, RRID:AB_10610636 | ICC (1:200), ChIP-qPCR, WB (1:2500) |
| Antibody | Anti-BRCA1 (Mouse monoclonal) | Santa Cruz Biotechnology | sc-6954, RRID:AB_626761 | ChIP-qPCR |
| Antibody | Anti-pATM (Ser1981) (Mouse monoclonal) | Cell Signaling | 4526, RRID:AB_2062663 | ChIP-qPCR |
| Antibody | Anti-γH2AX (Ser139) (Mouse monoclonal) | Millipore | 05-636, RRID:AB_309864 | ChIP-qPCR |
| Antibody | Anti-pATR (Thr1989) (Rabbit polyclonal) | Cell Signaling | 58014, RRID:AB_2722679 | ChIP-qPCR |
| Antibody | Anti-53BP1 (Rabbit polyclonal) | Santa Cruz Biotechnology | sc-22760, RRID:AB_2256326 | ChIP-qPCR |
| Antibody | Anti-Katushka2S (Rabbit polyclonal) | Eurogene | AB233, RRID:AB_2571743 | ChIP-seq |
| Antibody | Anti-GAPDH, (Rabbit polyclonal) | Abcam | ab9485, RRID:AB_307275 | WB |
| Antibody | Polyclonal Goat anti-mouse Alexa Fluor Plus 488 | Invitrogen | A32723, RRID:AB_2633275 | ICC (1:200) |
| Antibody | Polyclonal Goat anti-mouse Alexa Fluor Plus 594 | Invitrogen | A32742, RRID:AB_2762825 | ICC (1:200) |
| Antibody | Polyclonal Goat anti-rabbit CF488A | Biotium | #20012, RRID:AB_10853801 | ICC (1:200) |
| Antibody | Polyclonal Goat anti-rabbit CF594 | Biotium | #20112, RRID:AB_10559190 | ICC (1:200) |
| Commercial assay or kit | Duolink In Situ Detection Reagents Green | Sigma-Aldrich | DUO92014 | |
| Commercial assay or kit | Click-i RNA Alexa Fluor 488 Imaging Kit | Thermo Fisher | C10329 | |
| Software, algorithm | ImageJ (version 1.44) | https://imagej.net/ij/download.html | RRID:SCR_003070 | |
| Software, algorithm | CellProfiler (version 4.2.8) | https://cellprofiler.org/releases | RRID:SCR_007358 | |
| Software, algorithm | Bowtie (version 2.2.3) | https://bowtie-bio.sourceforge.net/bowtie2/index.shtml | RRID:SCR_016368 | |

*Continued on next page*

*Continued*

| Reagent type (species) or resource | Designation | Source or reference | Identifiers | Additional information |
|---|---|---|---|---|
| Software, algorithm | SAMtools (version 1.5) | https://github.com/samtools/samtools | RRID:SCR_002105 | |
| Software, algorithm | deepTools (version 3.4.2) | https://github.com/deeptools/deepTools | RRID:SCR_016366 | |
| Other | Lipofectamine LTX and Plus Reagent | Invitrogen | #15338-100 | |
| Other | Lipofectamine 3000 | Invitrogen | L3000-015 | |

Abbreviations: ICC, immunocytochemistry; ChIP, chromatin immunoprecipitation; WB: western blotting.

## Cell culture and drug treatment

Human HeLa (ATCC CCL-2), MCF7 (ATCC HTB-22), and human skin fibroblasts (female 46XX) were cultured in DMEM (PanEco) supplemented with 10% fetal bovine serum (FBS; HyClone/GE Healthcare) and penicillin/streptomycin. The cells were cultured at 37°C in a conventional humidified $CO_2$ incubator. DNA damage was induced by the treatment of cells with 90 μM etoposide (Sigma-Aldrich, #E1383) for 30 min or 1 hr. For rRNA transcription inhibition, cells were treated with 0.05 μg/ml AMD (Sigma-Aldrich, #A1410) or 20 μM CX-5461 (Sigma-Aldrich, #5092650001) for 3 hr. To obtain 1,6-HD-treated cells, HeLa cells were incubated with 5% 1,6-HD (Sigma-Aldrich, #240117) in serum-free medium at 37°C in a humidified atmosphere for 10 min. To obtain ammonium acetate-treated cells, HeLa cells were incubated with 200 mM ammonium acetate in a complete culture medium at room temperature for 5 min.

## Plasmid constructs

For the FusionRED-Treacle-Cry2 (opto-Treacle) construct, the full-length Treacle was amplified by PCR from cDNA with primer set #1 (*Supplementary file 1*) using KAPA High-Fidelity DNA Polymerase (KAPA Biosystems, KE2502). The forward and reverse primers contained XhoI sites. The amplified fragment was inserted into the FusionRed-C vector (Evrogen, FP411) linearized with XhoI. The Cry2 fragment was amplified by PCR from the plasmid pHR-mCh-Cry2WT (Addgene, #101221) with primer set #2 (*Supplementary file 1*) digested with NheI and inserted into pFusionRed-Treacle linearized with NheI using NheI enzyme.

For the FusionRED-FUS-Cry2 construct, the FUS was amplified by PCR from the plasmid pHR-FUSN-mCh-Cry2WT (Addgene, #101223) with primer set #3 (*Supplementary file 1*) using KAPA High-Fidelity DNA Polymerase (KAPA Biosystems, KE2502). The forward and reverse primers contained BspEI and KpnII sites respectively. The amplified fragment was inserted into the pFusionRed-C vector (Evrogen, FP411). The Cry2 fragment was amplified by PCR from the plasmid pHR-mCh-Cry2WT (Addgene, #101221) with primer set #2 (*Supplementary file 1*) digested with NheI and inserted into pFusionRed-Treacle using NheI enzyme.

To generate Treacle-GFP or Treacle-Katushka2S constructs, the full-length Treacle was amplified by PCR from cDNA with primer set #4 (*Supplementary file 1*) using KAPA High-Fidelity DNA Polymerase (KAPA Biosystems, KE2502). The forward and reverse primers contained BglII and BamHI sites respectively. The amplified fragment was inserted into the pTurboGFP-C (Evrogen, FP511) or pKatushka2S-C (Evrogen, FP761) vectors using BglII/BamHI restriction/ligation.

Treacle Δ1–83 deletion mutant was constructed based on the pTreacle-GFP full or pTreacle-2S full plasmid using iProof High-Fidelity DNA polymerase with primer set #5 (*Supplementary file 1*). The resulting DNA template after PCR was reprecipitated and treated with the DpnI restriction enzyme. In the next step, the desired DNA template was purified on an agarose gel, phosphorylated with T4 Polynucleotide Kinase (T4 PNK), and ligated.

To obtain the charge-scrambled Treacle-GFP or Treacle-2S (Treacle-GFP CS or Treacle-2S CS), a fragment encoding 137–1130 aa of Treacle was removed from the pTreacle-GFP or pTreacle-2S plasmid using EcoR1 and HindIII restrictases. This fragment was then replaced with a synthetic sequence encoding the amino acid sequence with the necessary changes (*Supplementary file 2*).

To obtain the Treacle ΔSE mutant, a fragment encoding 137–1130 aa of Treacle was removed from the pTreacle-GFP or pTreacle-2S plasmid using EcoR1 and HindIII restrictases. This fragment was then replaced with a synthetic sequence encoding the amino acid sequence with the necessary changes (*Supplementary file 2*).

Treacle Δ83–1121 deletion mutant was constructed based on the pTreacle-GFP full or pTreacle-2S full plasmid using iProof High-Fidelity DNA polymerase with primer set #6 (*Supplementary file 1*). The resulting DNA template after PCR was reprecipitated and treated with the DpnI restriction enzyme. In the next step, the desired DNA template was purified on an agarose gel, phosphorylated with T4 Polynucleotide Kinase (T4 PNK), and ligated.

Treacle Δ1121–1488-NLS mutant was constructed based on the pTreacle-GFP full or pTreacle-2S full plasmid using iProof High-Fidelity DNA polymerase with primer set #7 (*Supplementary file 1*). The resulting DNA template after PCR was reprecipitated and treated with the DpnI restriction enzyme. In the next step, the desired DNA template was purified on an agarose gel, phosphorylated with T4 Polynucleotide Kinase (T4 PNK), and ligated.

For the generation of Treacle ΔNoLS (Δ1350–1488) mutant, fragments were amplified by PCR from the pTreacle-GFP plasmid with primer set #8 (*Supplementary file 1*). The forward and reverse primers contained BglII and BamHI sites respectively. The amplified fragment was inserted into either the pTurboGFP-C (Evrogen, FP511) or pKatushka2S-C (Evrogen, FP761) vectors using BglII/BamHI restriction/ligation.

The full-length Treacle without fluorescent protein was constructed based on the pTreacle-2S full plasmid using iProof High-Fidelity DNA polymerase with primer set #9 (*Supplementary file 1*). The resulting DNA template after PCR was reprecipitated and treated with the DpnI restriction enzyme. In the next step, the desired DNA template was purified on an agarose gel, phosphorylated with T4 Polynucleotide Kinase (T4 PNK), and ligated.

Fragment of the wt Treacle (amino acids 291–426) were amplified by PCR with primer set #10 (*Supplementary file 1*) and subcloned into pMAL (New England Biolabs) vector encoding the TEV protease cleavage site after maltose-binding protein (MBP). cDNA fragments and the expression vector were cleaved with BamHI I and Sal I digestion and ligated under suitable conditions.

The resulting all of constructs were verified by sequencing.

## Gene knockdown

RNA interference experiments were performed using Lipofectamine 3000 transfection reagent (Thermo Scientific) following the manufacturer's instructions. The cells were transfected with 50 nM Treacle/TCOF1 siRNA (the sequences of the siRNA is provided in *Supplementary file 3*) or 50 nM TOPBP1 (Santa Cruz Biotechnology, #sc-41068).

For CRISPR/Cas9-mediated knockout, two sgRNA to first exon of the TCOF1 gene were designed using the guide RNA design tool (https://www.atum.bio/eCommerce/cas9/inputhttps://www.atum.bio/eCommerce/cas9/input). The sgRNA targeting sequences were separately cloned into the px330mCherry (Addgene #98750). A list of all oligonucleotides is provided in *Supplementary file 3*. The plasmids were co-transfected into HeLa cells with LTX transfection reagent (Invitrogen). 5–7 days after transfection, cells were fixed and immunostained with required antibodies.

## Fluorescence microscopy

For immunostaining, cells were grown on microscope slides. All samples were fixed in 1% formaldehyde in PBS for 15 min at room temperature and treated with 1% Triton X-100 for permeabilization. Cells were washed with PBS and then incubated with antibodies in PBS supplemented with 1% BSA and 0.05% Tween-20 for 1 hr at room temperature or overnight at 4°C. Then, the cells were washed with PBS three times (5 min each time). The primary antibodies bound to antigens were visualized using Alexa Fluor 488-conjugated secondary antibodies. The DNA was counterstained with the fluorescent dye 4,6-diamino-2-phenylindole (DAPI) for 10 min at room temperature. The samples were mounted using Dako fluorescent mounting medium (Life Technologies). The immunostained samples were analyzed using a Zeiss AxioScope A.1 fluorescence microscope (objectives: Zeiss N-Achroplan ×40/0.65 and EC Plan-Neofluar ×100/1.3 oil; camera: Zeiss AxioCam MRm; acquisition software: Zeiss AxioVision Rel. 4.8.2; Jena, Germany) or STELLARIS 5 Leica confocal microscope (objectives: HC PL APO ×63/1.40 oil CS2). The images were processed using ImageJ software (version 1.44) and analyzed using CellProfiler software (version 3.1.5). 3D reconstruction of xyz confocal datasets (z-stacks) was performed using Leica LAS-X software. Antibodies used in the study are listed in Key Resources Table.

## Cell sorting

Cells were transiently transfected with required plasmids using LTX transfection reagent (Invitrogen) or immunostained with required antibodies. Cell sorting was performed using an SH800 Cell Sorter (Sony) with a laser tuned to 488 nm for green fluorescence and 561 nm for red fluorescence. Gates were set with reference to negative controls. A minimum of $2 \times 10^6$ events was collected for ChIP or RNA extraction.

## RNA extraction and RT-qPCR

After sorting, living or immunostained cells were pelleted by centrifugation at 3000×$g$ for 5 min at 4°C. The supernatant was discarded. Total RNA was isolated from the pellet as described (*Hrvatin et al., 2014*). All RNA samples were further treated with DNase I (Thermo Scientific) to remove the residual DNA. RNA (1 µg) was reverse-transcribed in a total volume of 20 µl for 1 hr at 42°C using 0.4 µg of random hexamer primers and 200 U of reverse transcriptase (Thermo Scientific) in the presence of 20 U of ribonuclease inhibitor (Thermo Scientific). The cDNA obtained was analyzed by qPCR using the CFX96 real-time PCR detection system (Bio-Rad Laboratories). The PCRs were performed in 20 µl reaction volumes that included 50 mM Tris-HCl (pH 8.6), 50 mM KCl, 1.5 mM MgCl$_2$, 0.1% Tween-20, 0.5 µM of each primer, 0.2 mM of each dNTP, 0.6 µM EvaGreen (Biotium), 0.2 U of Hot Start Taq Polymerase (Sibenzyme), and 50 ng of cDNA. Primers used in the study are listed in *Supplementary file 4*.

## ChIP and ChIP-seq analysis

Living cells were fixed for 15 min with 1% formaldehyde at room temperature, and crosslinking was quenched by adding 125 mM glycine for 5 min. Cell sorting was performed if needed. Cells were harvested in PBS, and nuclei were prepared by incubation in FL buffer (5 mM PIPES, pH 8.0, 85 mM KCl, 0.5% NP-40) supplemented with Protease Inhibitor Cocktail (Bimake) and Phosphatase Inhibitor Cocktail (Bimake) for 30 min on ice. Next, chromatin was sonicated in RIPA buffer (10 mM Tris-HCl, pH 8.0, 140 mM NaCl, 1% Triton X-100, 0.1% sodium deoxycholate, 0.1% SDS) with a VirSonic 100 to an average length of 200–500 bp. Per ChIP reaction, 10–20 µg chromatin was incubated with 2–4 µg antibodies overnight at 4°C. The next day, Protein A/G Magnetic Beads (Thermo Scientific) were added to each sample and incubated for 4 hr at 4°C. Immobilized complexes were washed two times for 10 min at 4°C in low salt buffer (20 mM Tris-HCl, pH 8.0, 150 mM NaCl, 2 mM EDTA, 0.1% SDS, 1% Triton X-100) and high salt buffer (20 mM Tris-HCl, pH 8.0, 500 mM NaCl, 2 mM EDTA, 0.1% SDS, 1% Triton X-100). Samples were incubated with RNase A (Thermo Scientific) for 30 min at room temperature. The DNA was eluted from the beads and de-crosslinked by proteinase K digestion for 4 hr at 55°C and subsequent incubation at 65°C for 12 hr. Next, DNA was purified using phenol/chloroform extraction and analyzed by qPCR. The qPCR primers used for ChIP analysis are listed in *Supplementary file 5*. The sequencing libraries were then prepared with NEBNext Ultra II kit according to the manufacturer's protocol. Final libraries were PCR amplified and adapter dimers were cleaned with 1:1 MagPure magnetic beads (Magen Biotechnology). Resulted DNA was resuspended in 30 µl 10 mM Tris-HCl buffer pH 8.0 and were sequenced on Illumina machine. ChIP-seq reads were mapped to the reference human genome hg38 assembly using Bowtie v2.2.3 with the '–very-sensitive' mode. Non-uniquely mapped reads, possible PCR, and optical duplicates were filtered out using SAMtools v1.5. The bigWig files with the ratio of RPKM normalized ChIP-seq signal to the input were generated using deepTools v3.4.2 bamCompare function.

## Electron microscopy

Twenty-four hours after transfection, cells were fixed with 2.5% neutralized glutaraldehyde in the requisite buffer for 2 hr at room temperature, post-fixed with 1% aqueous OsO$_4$, and embedded in Epon. Sections of 100 nm thickness were cut and counterstained with uranyl acetate and lead citrate. Sections were examined and photographed with a JEM 1400 transmission electron microscope (JEOL, Japan) equipped with a QUEMESA bottom-mounted CCD camera (Olympus SIS, Japan) and operated at 100 kV.

## Protein purification

*E. coli* BL21 (DE3) cells were transformed with vectors encoding cDNAs of TCOF LCR fragments fused with TEV-cleavable MBP. BL21 (DE3) were cultured in Terrific Broth medium at 37°C to OD=0.6–1.0.

Protein expression was induced by adding 0.8 mM IPTG, and the cultures were further incubated at 18°C overnight. The cells were collected by centrifugation (5000×g, 20 min, 4°C). To purify MBP-tagged protein under a native condition, the cell pellet was dissolved in start buffer (20 mM Tris-HCl, 150 mM NaCl, 10 mM MgCl₂, 0.1% NP-40, 10% glycerol, 1 mM DTT, 1 mM phenylmethylsulfonyl fluoride, 1× protease inhibitor cocktail [B14001 selleckchem], pH 7.4). Cells were disrupted by sonication in start buffer on ice. The lysate was centrifuged (20,000×g, 4°C, 60 min) and the supernatant was collected and applied to a column containing amylose resin (NEB, E8021S) at 4°C. The column was washed with wash buffer (20 mM Tris-HCl, 500 mM NaCl, 10 mM MgCl₂, 0.1% NP-40, 10% glycerol, 1 mM DTT, pH 7.4).

For cleavage of MBP TEV protease was added at molar ratio approximately 1:50 directly to the eluted protein in buffer (20 mM Tris-HCl, 200 mM NaCl, 5 mM sodium citrate, 1 mM DTT, pH 7.4) overnight at 4°C while gently mixing the resin on a rotator. Subsequently, the tagged proteins were eluted and then subjected to ion-exchange chromatography with 0–1000 mM NaCl gradient.

Eluted proteins were sequentially dialysed at 4°C against assay buffer (20 mM Tris-HCl pH 7.4) at 4°C overnight and concentrated using Amicon centrifugal filters (Millipore) and stored at −80°C in small aliquots.

### In vitro Treacle condensation analysis

Recombinant Treacle fragment (amino acids 291–426) was diluted in 20 mM Tris-HCl pH 7.4 to a final concentration of 100 µM. To assemble condensates PEG 8000 was added to the protein solution to a final concentration of 5%, followed by incubation at room temperature for 10 min. 10 µl of the resulting solution was pipetted onto a glass slide, covered with a coverslip, and analyzed on the Zeiss AxioScope A.1 fluorescence microscope in DIC mode. To test the effects of NaCl or 1,6-HD on Treacle condensation, NaCl or 1,6-HD was added to the protein solution with 5% PEG 8000 at 500 mM or 10% concentrations, respectively.

### Live-cell imaging

Cells were seeded in 35 mm glass-bottom dishes 12 hr before transfection with the plasmid. Twenty-four hours after transfection, Hoechst 33342 (Cell Signaling Technology) was added to the medium at a final concentration of 1 µg/ml, and the cells were incubated for 20 min at 37°C. Hoechst-containing medium was replaced with a complete fresh medium before the cells underwent live-cell imaging using a STELLARIS 5 Leica confocal microscope (objectives: HC PL APO ×63/1.40 oil CS2) equipped with an incubation chamber to provide a humidified atmosphere at 37°C with 5% CO₂. For observation of liquid droplet behavior, z-stack time-lapses were taken. Maximum intensity projections are shown, but all findings were confirmed on a single plane.

### optoDroplet assay

Cells were seeded in 35 mm glass-bottom dishes and were transfected with Treacle-FusionRed-Cry2 plasmid using LTX transfection reagent (Invitrogen) 24 hr before imaging. Twenty-four hours after transfection, Hoechst 33342 was added to the medium at a final concentration of 1 µg/ml, and the cells were incubated for 10 min at 37°C. Hoechst-containing medium was replaced with a complete fresh medium before the cells underwent live-cell imaging using a STELLARIS 5 Leica confocal microscope (objectives: HC PL APO ×63/1.40 oil CS2) equipped with an incubation chamber to provide a humidified atmosphere at 37°C with 5% CO₂. Droplet formation was induced with light pulses at 488 nm (blue light, 1% laser power) for 10 s, and z-stack images were captured every 60 s in the absence of blue light.

### FRAP analysis

FRAP experiments were performed on a STELLARIS 5 Leica confocal microscope equipped with an HC PL APO ×63/1.40 oil CS2 objective and an incubation chamber that maintained a humidified atmosphere at 37°C with 5% CO₂. Live-cell experiments were performed using transfected cells cultured in phenol red-free DMEM medium supplemented with 10% fetal bovine serum. Imaging was typically performed at a resolution of 512 × 512 pixels, with a scan speed of 200 ms per frame.

Typically, images were acquired at 512 × 512 pixels at a scan speed corresponding to 200 ms per image. Before photobleaching, three to five images were recorded. Bleaching parameters, i.e., laser

intensity and scanning time, were chosen to reach approximate 50% of bleaching in the shortest possible time. The bleaching area was selected according to the type of experiment (full-, partial-, or half-FRAP). In the case of half-FRAP, which relies on the analysis of the bleached and the non-bleached half, it is important to optimize the bleaching step (intensity and area of the bleaching spot) to minimize bleaching in the other (non-bleached) half. Half-FRAP experiments were conducted until the signals in both halves had converged to each other or until the signal in the non-bleached half had reached its initial pre-bleach value. FRAP data analysis was performed as described in *Muzzopappa et al., 2022*.

## Proximity ligation assay

After drug treatments or at 24 hr post-transfection, cells were fixed for 15 min with 1% formaldehyde at room temperature and permeabilized with 1% Triton X-100. Next, cells were subjected to the PLA using the Duolink Green (Sigma-Aldrich) according to the manufacturer's instructions. Briefly, cells were blocked, incubated with appropriate primary antibodies overnight at 4°C, and then incubated with PLA probes, which are secondary antibodies (anti-mouse-IgG and anti-rabbit-IgG) conjugated to unique oligonucleotides. Samples were then treated with a ligation solution followed by an amplification solution containing polymerase and fluorescently labeled oligonucleotides, allowing rolling-circle amplification and detection of discrete fluorescent dots. Some samples after the PLA protocol were additionally counterstained with Alexa Fluor 488- or Alexa Fluor 594-conjugated (Molecular Probes) secondary antibody. The DNA was counterstained with DAPI for 10 min at room temperature. The samples were mounted using Dako fluorescent mounting medium (Life Technologies) and analyzed using STELLARIS 5 Leica confocal microscope (objectives: HC PL APO ×63/1.40 oil CS2). The images were analyzed using ImageJ software (version 1.44).

## Transcription labeling

For EU incorporation, the cells were incubated with 100 μM EU (Sigma-Aldrich) for 2 hr at 37°C. Then, the cells were washed three times with PBS and fixed in ice-cold methanol for 10 min. The samples were then processed using a Click-iT EU Imaging Kit (Life Technologies) according to the manufacturer's recommendations.

## Single-molecule RNA FISH

All single-molecule RNA FISH probes were designed as described (*Yao et al., 2019*) and labeled with FITC on the 3' ends (*Supplementary file 6*). Cells were fixed with 4% PFA for 15 min, followed by permeabilization with 1% Triton X-100 for 10 min. Cells were incubated in 10% formamide/2× SSC for 10 min at room temperature and were then hybridized with 5 nM each of RNA probes in 50% formamide/2× SSC at 37°C for 16 hr. After hybridization, the cells were washed in 10% formamide/2× SSC for 30 min at 37°C.

## Acknowledgements

This work was supported by RSF grant 21-74-10018. Confocal microscopy studies were supported by grant 075-15-2019-1661 from the Ministry of Science and Higher Education of the Russian Federation.

## Additional information

### Funding

| Funder | Grant reference number | Author |
|---|---|---|
| Russian Science Foundation | grant 21-74-10018 | Artem K Velichko |
| Ministry of Science and Higher Education of the Russian Federation | grant 075-15-2019-1661 | Sergey Razin |

| Funder | Grant reference number | Author |
|--------|------------------------|--------|

The funders had no role in study design, data collection and interpretation, or the decision to submit the work for publication.

## Author contributions
Artem K Velichko, Conceptualization, Investigation, Methodology, Writing – original draft; Nadezhda V Petrova, Dmitry A Deriglazov, Anastasia P Kovina, Investigation, Methodology; Artem V Luzhin, Formal analysis, Validation, Investigation, Methodology; Eugene P Kazakov, Igor I Kireev, Methodology; Sergey Razin, Conceptualization, Resources, Supervision, Funding acquisition, Project administration, Writing - review and editing; Omar L Kantidze, Conceptualization, Supervision

## Author ORCIDs
Artem K Velichko ⬡ http://orcid.org/0009-0006-3105-1988
Omar L Kantidze ⬡ https://orcid.org/0000-0002-7507-7307

Reviewer #1 (Public review): https://doi.org/10.7554/eLife.96722.3.sa1
Reviewer #2 (Public review): https://doi.org/10.7554/eLife.96722.3.sa2
Reviewer #3 (Public review): https://doi.org/10.7554/eLife.96722.3.sa3
Author response https://doi.org/10.7554/eLife.96722.3.sa4

# Additional files

## Supplementary files
Supplementary file 1. List of primers used for cloning.

Supplementary file 2. The amino acid substitutions and deletions engineered to generate the Treacle ΔSE and Treacle CS mutant variants.

Supplementary file 3. List of primers used for knockdown and knockout.

Supplementary file 4. List of primers used for quantitative polymerase chain reaction (qPCR).

Supplementary file 5. List of primers used for chromatin immunoprecipitation (ChIP)-quantitative polymerase chain reaction (qPCR).

Supplementary file 6. List FITS-labeled oligonucleotides used for single-molecule fluorescence in situ hybridization (smFISH).

MDAR checklist

## Data availability
The ChIP-seq data was submitted to GEO. The GEO Accession number is GSE292850.

The following dataset was generated:

| Author(s) | Year | Dataset title | Dataset URL | Database and Identifier |
|-----------|------|---------------|-------------|--------------------------|
| Velichko AK, Luzhin AV, Razin SV | 2025 | Treacle's ability to form liquid-like phase condensates is essential for nucleolar fibrillar center assembly, efficient rRNA transcription and processing, and rRNA gene repair | https://www.ncbi.nlm.nih.gov/geo/query/acc.cgi?acc=GSE292850 | NCBI Gene Expression Omnibus, GSE292850 |

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
