## [Editor Report · eLife Assessment]

This **important** study reveals that the nucleolar protein Treacle undergoes liquid-liquid phase separation in vitro and in vivo. It provides **convincing** evidence that the ability of Treacle to form phase-separated condensates is necessary for the proper formation of the fibrillar center of the nucleolus, rRNA transcription, and rDNA repair. These findings will be of interest to the communities studying biomolecular condensates, nucleolar organization, and ribosome biogenesis.

---

## [Referee Report · Reviewer #1 (Public review)]

Summary:

The manuscript by Velichko et al. argues that the ability of nucleolar protein Treacles to form phase-separated condensates is necessary for its function in nucleolar organization, rRNA transcription, and rDNA repair. These findings may be of interest to the communities studying biomolecular condensates, nucleolar organization, and ribosome biogenesis. The authors propose that Treacle's ability to undergo liquid-liquid phase separation is the key to its role as a scaffold for the FC of the nucleolus. The experiments in this study were designed and performed well, particularly the overexpression studies, done in the absence of endogenous protein and accounted for the protein expression levels.

Comments on revisions:

I am satisfied with the authors' revisions; my earlier concerns have been addressed thoroughly, and the manuscript is considerably improved. This study is important for our understanding of the role of Treacle in nucleolar organization and function, as well as general principles of cellular compartmentalization that involve biomolecular condensates.

---

## [Referee Report · Reviewer #2 (Public review)]

Summary:

Velichko, et al. investigate the role played by the long intrinsically disordered protein Trecle in nucleolar morphology and function, with an interest in its potential ability to undergo condensation. The authors explore Treacle's role in core functions of the nucleolus (rRNA biogenesis and DNA repair), which has been a subject of continual investigation since it was identified that truncation of Treacle is the primary genetic cause of Treacher-Collins syndrome. They show that knock out of Treacle leads to de-mixing of canonical markers of the FC (UBF, RPA194) and DFC (FBL) phases of the nucleolus. They also show that replacing Treacle with mutants that either remove the central region of Treacle (∆83-1121) or reduce the segregation of charged residues by scrambling them (CS- Charge Scrambled) results in different FRAP behavior of the condensates that result from Treacle over-expression. These data give new insight into the role played by the charge-segregated central region of Treacle in terms of having the potential to undergo condensation.

Strengths:

The characterizations of changes to nuclear morphology upon Treacle knockout is the strength of this study. The authors characterized effects on the canonical markers of the FC and DFC phases support the idea that Treacle has a scaffolding function. While the effect of Treacle perturbations has been studied before, this has often been investigated in the context of organismal development or rRNA biogenesis and less often at the sub-cellular level, as the authors have carried out.

Another strength of this study is its characterization of the effects of the charge scramble mutant. The authors find that replacing endogenous Treacle with this mutant reduces the bulk dynamics of Treacle as assessed by FRAP, de-mixes FBL from the DFC, lowers pre-rRNA synthesis, and abolishes the recruitment of the DNA-damage response factor TOPBP1.

Weaknesses:

The conclusion that Treacle is a core scaffold of the FC is weakly supported. Recombinant Treacle has intrinsic potential to condense, and its condensation is disrupted by the expected solution conditions (i.e., condensates fail to form at high salt but do form in the presence of an aliphatic alcohol). It should be kept in mind that all proteins will condense at sufficiently high concentrations and under crowding. The authors observed condensation at 100uM protein and 5% PEG8000.

---

## [Referee Report · Reviewer #3 (Public review)]

Summary:

This study provides evidence that the protein Treacle plays an essential role in the structure and function of the fibrillar center (FC) of the nucleolus, which is surrounded by the dense fibrillar component (DFC) and the granular component (GC). The authors provide new evidence that, like the DFC and GC, the functional FC compartment involves a biomolecular condensate that contains Treacle as a key component. Treacle is essential to transcription of the rDNA as well as proper rRNA processing that the authors tie to a role in maintaining separation of FC components from the DFC. In vitro and in vivo experiments highlight that Treacle is itself capable of undergoing condensation in a manner that depends on concentration and charge-charge interactions, but is not affected by 1,6 hexanediol, which disrupts weak hydrophobic interactions. Attempting to generate separation-of-function mutants, the authors provide further evidence of complex interactions that drive proper condensation in the FC mediated by both the central repeat (low-complexity, likely driving the condensation) and C-terminal domain (which appears to target the specificity of the condensation to the proper location). Using mutant forms of Treacle defective in condensation, the authors provide evidence that these same protein forms are also disrupted in supporting Treacle's functions in rDNA transcription and rRNA processing. Last, the authors suggest that cells lacking Treacle are defective in the DNA damage response at the rDNA in response to VP16.

Strengths:

In general, the data are of high quality, the experiments are well-designed and the findings are carefully interpreted. The findings of the work complement prior high-impact studies of the DFC and GC that have identified constituent proteins as the lynchpins of the biomolecular condensates that organize the nucleolus into its canonical three concentric compartment structure and are therefore likely to be of broad interest. The attempts to generate separation-of-function mutants to dissect the contribution of condensation to Treacle function are ambitious and critical to demonstrating the relevance of this property to the biology of the FC. The complementarity of the methods applied to investigate Treacle function are appropriate and the findings integrate well towards a compelling narrative.

Weaknesses:

While the separation of function mutants of Treacle are a major strength of the work, further studies will be required to fully explore the relevance of Treacle condensation to the stability of the rDNA repeats.

---

## [Author Response]

The following is the authors’ response to the original reviews

**Recommendations for the authors:**

**Reviewer #1 (Recommendations for the Authors):**
The interpretation of results obtained with opto-Treacle (related to Figure 2C) may be expanded.

We thank the reviewer for their insightful comment regarding the interpretation of the results obtained with opto-Treacle. We understand the concern that the difference in the size of the condensates formed by opto-Treacle (Figure 2C) compared to Treacle-2S or other constructs may raise questions about the role of tetramerization in driving condensate formation, as 2S is known to tetramerize while FusionRed is not susceptible to multimerization.

To address this concern, we emphasize that we have demonstrated that overexpressed Treacle forms large condensates even in the absence of any fluorescent protein, as included in the revised manuscript. This observation supports the conclusion that Treacle's ability to form condensates is intrinsic and does not depend on the multimerization capacity of the fluorescent tag.

We believe that the observed difference in condensate size between opto-Treacle and Treacle-2S, Treacle-GFP, or untagged Treacle arises primarily from the time available for condensate assembly. Opto-Treacle condensation occurs rapidly, within approximately 10 seconds of blue light illumination, whereas Treacle-2S, Treacle-GFP, or untagged Treacle undergo condensation over the extended period of 24–48 hours of protein overexpression. This temporal difference likely accounts for the disparity in condensate size, as longer assembly times allow for larger and more mature condensates to form.

Given this reasoning, we consider it unnecessary to further emphasize the size differences in the main text of the article, as we believe the underlying explanation is clear and supported by the data. Nonetheless, we are open to incorporating additional clarifications if the reviewer deems it necessary.

The authors might reconsider referring to Treacle as a scaffold. Ultimately, the scaffold for the nucleolus is the rDNA with its bound proteins. Scaffold proteins, by definition, bind multiple protein partners and facilitate the formation of multiprotein complexes, a role not really attributed to homotypic LLPS.

We thank the reviewer for raising this important point regarding the use of the term "scaffold" in relation to Treacle. We fully acknowledge that rDNA, along with its associated protein complexes, serves as the primary structural scaffold for the nucleolus. However, we believe that referring to Treacle as a scaffold is appropriate and justified within the specific context of our study.

First, we emphasize that we describe Treacle as a scaffold specifically for nucleolar fibrillar centers (FCs), rather than for the nucleolus as a whole. This distinction is important, as our work focuses on the role of Treacle in organizing FC components, rather than the broader structural organization of the nucleolus.

Second, as the reviewer notes, scaffold proteins are defined by their ability to bind multiple protein partners and facilitate the formation of multiprotein complexes. Our findings demonstrate that Treacle's condensation properties promote the binding and retention of key rDNA-associated protein partners, including RPA194, UBF, and Fibrillarin, within the FCs. This activity aligns with the functional definition of a scaffold protein, as Treacle supports the spatial organization and cooperative interactions of FC components essential for rRNA transcription and processing. Therefore, while we appreciate the reviewer's observation regarding the central role of rDNA as a nucleolar scaffold, we maintain that the use of the term "scaffold" to describe Treacle's role in organizing FCs is consistent with its demonstrated functional properties.

If authors decide to add the "Ideas and Speculation" subsection to their Discussion, it may be interesting to discuss the following outstanding questions: does Treacle undergo homotypic or heterotypic LLPS? Does its overexpression favor homotypic interactions? How does it segregate FC and DFC compartments -by exclusion? How does phase-separated Treacle interact with other proteins?

We thank the reviewer for these insightful questions. While we believe that adding a dedicated "Ideas and Speculation" subsection would be redundant, we have already addressed the questions regarding Treacle’s homotypic or heterotypic LLPS and its interactions with other proteins in the revised "Discussion" section. Additionally, we have included a new section in the manuscript specifically focused on investigating the role of Treacle condensation in its interactions with protein partners, further expanding on these points.

In Materials and Methods, smFISH section -"probes were designed as described (Yao et al, 2019) and labeled with FITS on the 3'ends" - was it meant to say FITC (i.e. Fluorescein)?

We thank the reviewer for catching this error. This was indeed a typo, and we have corrected it to "FITC (i.e., Fluorescein)" in the revised text.

**Reviewer #2 (Recommendations for the Authors):**
Regarding recombinant Treacle, the main concern is that the authors may not be observing the condensation of Treacle itself. The quality of the purchased recombinant Treacle is unclear (this reviewer could not find Treacle listed on the vendor website despite using the supplied catalog number or vapors search terms). Furthermore, it is not clear if the condensates observed are Treacle or potentially the Dextran crowder. Only small percentages (>1%-5%) of either Dextran or PEG are needed to induce phase separation in two-component mixtures of these polymers. PEG may be in the Treacle storage butter. In addition to clarifying the State of recombinant Treacle, these concerns could be further assuaged by direct visualizing of Treacle forming condensates (via fluorescent n-terminal tagging) and filling in more of the phase space to observe the loss of condensates at a threshold concentration of Treacle. In general, the gold standard for establishing condensation of a given protein is mapping the full binodal phase diagram diagram of the protein. Understanding that protein is a limited resource, most groups simply map the lower concentration arm of the binodal, and this is sufficient to characterize a protein as having intrinsic condensation behavior. A similar mapping effort of Treacle would be welcomed.

We thank the reviewer for their thoughtful comments and for highlighting concerns regarding the interpretation of our experiments with commercial recombinant Treacle. We recognize the importance of ensuring that the observed condensation properties are intrinsic to Treacle and not influenced by potential contaminants, storage buffer components, or tags on the protein.

To address these concerns, we have re-evaluated the condensation properties of Treacle using a recombinant fragment independently purified in our laboratory. Specifically, we expressed and purified a Treacle fragment (amino acids 291–426), which includes two S/E-rich low-complexity regions (LCRs) and two linker regions, in *E. coli*. The protein was expressed as a TEV-cleavable maltose-binding protein (MBP) fusion, purified under native conditions via amylose resin, and subjected to TEV cleavage. This was followed by ion-exchange chromatography and extensive dialysis to remove any remaining impurities. These additional steps ensured that the purified Treacle fragment was of high purity and free from confounding components, such as polyethylene glycol (PEG). We have included detailed descriptions of this protocol in the revised manuscript.

Using this purified Treacle fragment, we confirmed its intrinsic condensation behavior in vitro. In the presence of 5% PEG8000 as a crowding agent, the fragment formed liquid-like condensates that exhibited spherical morphology and dynamic fusion events, key hallmarks of liquid-liquid phase separation (LLPS). Additionally, we demonstrated that the condensation of this Treacle fragment was sensitive to changes in pH and salt concentration but unaffected by 1,6-hexanediol treatment, suggesting that the condensates are stabilized predominantly by electrostatic interactions (Fig. 4B of the revised manuscript). Importantly, these findings provide robust evidence that Treacle possesses intrinsic phase-separation properties. All results from the commercial Treacle protein used in the initial version of the manuscript have been replaced with data obtained using this independently purified recombinant fragment.

We undestand that the condensation behavior of the fragment may not fully capture the behavior of full-length Treacle. Nevertheless, the in vitro experiments provide valuable mechanistic insights into the biophysical properties of Treacle. Furthermore, as emphasized in the revised manuscript, our study primarily focuses on understanding the condensation and functional role of Treacle in a cellular context, where we observe its critical involvement in organizing nucleolar structure and regulating rRNA transcription. These cellular experiments highlight the biological relevance of Treacle’s condensation behavior.

With regard to mapping the binodal phase diagram of Treacle, we concur with the reviewer that such an effort would be ideal for a more comprehensive characterization of Treacle’s condensation properties. However, the limited availability of purified protein currently precludes a detailed mapping effort. Despite this limitation, we believe the qualitative assessments of Treacle’s condensation under varying conditions, now included in the revised manuscript, sufficiently demonstrate its intrinsic ability to phase-separate.

In conclusion, we are grateful for the reviewer’s feedback, which has allowed us to refine our methodology and strengthen the evidence supporting the intrinsic condensation properties of Treacle. We are confident that the revised manuscript provides a robust and thorough characterization of Treacle’s phase-separation behavior and its functional role in the cell, addressing the reviewer’s concerns. Thank you for your constructive recommendations, which have significantly improved the quality of our work.

Replacing 'liquid-phase' and 'liquid' with 'liquid-like' would make the language consistent with other papers in the field and more accurately reflect the degree of material state analysis carried out in the study.

We thank the reviewer for this insightful recommendation. In response to the suggestion, we have revised the manuscript to replace the terms "liquid-phase" and "liquid" with "liquid-like" throughout the text. This change ensures consistency with terminology commonly used in the field and more accurately reflects the degree of material state analysis performed in our study. We believe this adjustment improves the clarity and precision of our findings, aligning the manuscript with standard practices in the field. Thank you for helping us enhance the quality of the presentation.

The 'unclear' nature of the condensation behavior of the FC phase of the nucleolus is listed as a motivation for carrying out the study in the introduction; the authors could note here two recent papers that have investigated the nature of FC condensation: Jaberi-Lashkari et al. 2023 and King et al. 2024. The reviewer notes that while these were both pre-printed in late 2022, they were only recently published.

We thank the reviewer for bringing these recent studies to our attention. In response to the suggestion, we have cited the papers by Jaberi-Lashkari et al. (2023) and King et al. (2024) in both the introduction and discussion sections of the revised manuscript. These references are highly relevant to the context of our study and provide valuable insights into the condensation behavior of the FC phase of the nucleolus. We agree that incorporating these works strengthens the framing of our study and situates it more effectively within the broader field. Thank you for this constructive recommendation.

The statement that Treacle is "the main molecule present in the FC" is a substantial claim that does not need to be made to promote the author's case, nor is it well supported by the provided reference (Gal et al., 2022).

We thank the reviewer for pointing out this overstatement in our original manuscript. In response, we have revised the text to provide a more accurate and well-supported description. Specifically, we have replaced the claim that Treacle is "the main molecule present in the FC" with a statement highlighting its direct interactions with UBF and RNA Pol I, as well as its colocalization with these proteins within the FC. This revision ensures alignment with the provided references and more accurately reflects the current understanding of Treacle's role in the FC. We appreciate the reviewer's attention to this detail, which has helped us improve the clarity and accuracy of our manuscript.

The statement that "Treacle is one of the most intrinsically disordered proteins" is vague and unnecessarily grand. Treacle is a fully intrinsically disordered protein; these comprise 5% of the human proteome (Tsang et al. 2020), so Treacle is, indeed, unusual in that regard.

We thank the reviewer for highlighting the vague and unnecessarily broad nature of the original statement. In response, we have revised the text to provide a more precise and accurate description of Treacle's structural properties. Specifically, we replaced the claim that "Treacle is one of the most intrinsically disordered proteins" with the statement that "According to protein structure predictors (e.g., AlphaFold, IUPred2, PONDR, and FuzDrop), Treacle is a fully intrinsically disordered protein." This wording reflects the unique nature of Treacle while remaining scientifically accurate and supported by reliable computational predictions. We appreciate the reviewer's feedback, which has allowed us to improve the rigor and clarity of our manuscript.

A comment on the implications of the immobile pool of Treacle (which appears to be ~50% in WT and across a range of mutants) would be welcome. Additionally, the limitations of FRAP for interrogating material properties of condensed material in living systems are provided in Goetz and Mahamid, 2020. In this paper, the authors review instances where the ultrastructure of condensate is known and where FRAP data is available. They show that crystalline assemblies can recover faster than apparently liquid, spherical assemblies. A comment in the text about how these limitations apply to this study would be welcome.

We appreciate the reviewer’s insightful comments regarding the interpretation of the immobile pool of Treacle and the limitations of FRAP for characterizing material properties in living systems. As noted in our response to the public review, we believe the ~50% recovery rate after photobleaching observed in our experiments is best explained by the redistribution of Treacle molecules within the condensate, rather than significant exchange with the surrounding phase. This interpretation is strongly supported by the full- and half-FRAP analyses included in the revised manuscript, which demonstrated internal mixing dynamics within the condensates.

There appears to be a typo in the following sentence: "The highly positively charged CD serves as the nucleation center for RD but exhibits ambivalent phase properties, transitioning from LLPS to LSPS in the absence of rRNA." The LLPS to LSPS behavior was observed for mutants to the central domain (RD), not the c-terminal domain (CD).Throughout the authors report single snapshots of representative cells and single line traces. Analysis of the key morphological feature across the population of cells would help the reader understand how widespread the observed phenotype is.

We thank the reviewer for raising this important point regarding the representation of morphological features across the cell population. To address this concern, we have included widefield micrographs of cell fields in the revised figures to provide a more comprehensive view of the phenotypes observed.

The statement that "The phase behavior of polymers is determined by interactions through associative motifs, referred to as stickers, separated by spacers, which are not the primary driving forces for phase separation" could be improved by pointing out that this is potentially incomplete for describing the kind of condensation that highly charged polymers undergo. The high charge and charge segregation of Treacle suggest that it is a blocky polyampholyte and that it condenses by coacervation. Models of associative polymers can be useful for describing coacervation, however, the driving forces for coacervation are less understood and have been proposed to include an entropic component (see Sathyavageeswaran et al. 2024, Sing and Perry 2020 and work from their groups as well as the Obermayer (Columbia) and Terrell (U. Chicago) Groups).

We thank the reviewer for highlighting this important aspect of the phase behavior of charged polymers and for suggesting relevant references. In response, we have revised the discussion section of the manuscript to include a more nuanced explanation of the condensation mechanisms for highly charged polymers such as Treacle. Specifically, we now describe Treacle as a blocky polyampholyte, suggesting that its condensation behavior may be driven by coacervation mechanisms.The relevant references have been added to the discussion section of the revised manuscript.

In addition to the above, the authors may consider citing two recent publications from the Pappu group (King et al. Cell 2024 and King et al. Nucleus 2024) that directly investigate the condensation potential of K-rich and E/D-rich' grammars' on nucleolar proteins and show that, like the authors, the K-rich region is essential for localization and is conserved across nucleolar proteins.

We thank the reviewer for bringing these relevant publications to our attention. The suggested references from the Pappu group (King et al., Cell 2024, and King et al., Nucleus 2024) have been added to the introduction and discussion sections of the revised manuscript, and their findings have been appropriately integrated into our analysis.

The authors could consider replacing the use of LLPS with a more generic term such as "condensation" or "biomolecular condensation." LLPS of polymers is a segregative transition driven by its incompatibility with the surrounding solvent. As indicated, Treacle is likely to be undergoing some form of coacervation (which is predominantly an associative tradition), which can be genetically described as condensation. See Pappu et al. 2023 for more details.

We thank the reviewer for their insightful suggestion. Following the reviewer's recommendation, we have replaced the term "LLPS" with "condensation" or "coacervation" throughout the manuscript, where appropriate. Additionally, we have referenced Pappu et al. (2023) and other to provide further context and clarity regarding the distinctions between these terms.

The authors cite Yao et al. 2019, but do not cite the follow-up study (Wu et al. 2021) or provide a statement on how the Chan group finds a role for the RGG domain of FBL in keeping the certain canonical markers of the FC and DFC de-mixed.

We thank the reviewer for pointing out these important references. The relevant citations, including Wu et al. (2021), have been added to the manuscript.

**Reviewer #3 (Recommendations for the Authors):**
The following comment is true but could be broadened to include examples of structured regions promoting biomolecular condensation. "In biological systems, phase separation is mainly a characteristic of multivalent or intrinsically disordered proteins (Banani et al, 2017; Shin & Brangwynne,2017; Uversky, 2019)."

We have expanded the statement as recommended by the reviewer: "In biological systems, phase separation is facilitated by a combination of multivalent interactions mediated by intrinsically disordered proteins and site-specific interactions that drive percolation."

Related to Figure 1.The authors report Treacle-dependent EU incorporation (Figure 1D), but are there any changes more broadly to nucleolar number or size as a consequence? How do the authors interpret that the quantitative effect of AMD treatment is more extreme than Treacle depletion (Figure 1E).

We thank the reviewer for raising these important points. Regarding nucleolar number and morphology, we did not observe a change in the number of nucleoli upon Treacle depletion. However, nucleoli appeared more regularly rounded under these conditions, which we interpret as a consequence of the decreased rDNA transcription activity caused by Treacle depletion. A similar rounding of nucleoli is also observed upon actinomycin D (AMD) treatment, which is consistent with reduced transcriptional activity.

As for the more pronounced effect of AMD compared to Treacle depletion on EU incorporation, this can be explained by the fundamentally different mechanisms through which these conditions affect transcription. Treacle depletion reduces the local concentration of transcription factors at rDNA sites, thereby impairing transcription initiation and elongation to a certain extent. However, under Treacle depletion, RNA polymerase I still retains the ability to bind to the promoter and support a residual level of transcription. In contrast, AMD acts as a potent intercalator in GC-rich regions of rDNA, physically blocking the ability of RNA polymerase I to move along rDNA, resulting in near-complete cessation of rRNA synthesis.

Related to Figure 2.The authors observe that AMD leads to coalescence of individual Treacle-2S + bodies (e.g. Figure 2E) - does this suggest that ongoing rRNA transcription is required to prevent such events?

Thank you for your thoughtful question. Indeed, our observations strongly suggest that ongoing rRNA transcription is required to prevent the coalescence of Treacle-2S + bodies, as observed upon AMD treatment. This interpretation aligns with the findings of Tetsuya Yamamoto et al., who demonstrated that nascent ribosomal RNA (pre-rRNA) acts as a surfactant to suppress the growth and fusion of fibrillar centers (FCs) in the nucleolus. Their work highlighted that nucleolar condensates formed via liquid-liquid phase separation (LLPS) tend to grow to minimize surface energy, provided sufficient components are available. However, the transcription of prerRNA stabilizes FCs by maintaining multiple microphases, preventing coalescence unless transcription is inhibited.

According to Yamamoto et al., nascent pre-rRNAs tethered to FC surfaces by RNA Polymerase I generate lateral pressure that counteracts interfacial tensions, effectively suppressing FC fusion. This activity is analogous to the surfactant properties of molecules in physical systems. When transcription is inhibited (e.g., by AMD), the loss of nascent rRNA allows condensates to coalesce, consistent with the behavior we observe.

We further propose that the AMD-induced coalescence of Treacle-2S + bodies reflects the loss of this surfactant-like effect, as transcriptional activity ceases. This theory is also supported by the observation that Treacle condensates in the nucleoplasm, where rRNA transcription is absent, form larger structures. Collectively, these insights highlight the critical role of ongoing rRNA transcription in maintaining the structural integrity and dynamic organization of nucleolar substructures.

Related to Figure 3.In the figure panels B-H the DAPI signal in gray obscures the Treacle localization, especially in Figure 3H. A non-merged image for each of these examples for the Treacle localization would be very helpful.

We thank the reviewer for this observation. To address this, we have included wide-field images without the DAPI overlay for the deletion mutant lacking the 1121-1488 region. These are now presented in Supplementary Figure S5G of the revised manuscript.

Related to Figure 5.Only a single representative nucleus is shown in the PLA analysis presented in Figure 5B.Quantification to assess the robustness of this response with the addition of VP16 is needed. The authors use ChIP and immunocytochemistry as orthogonal methods but it would be best to therefore show both for each manipulation that is performed - the immunostaining of TOPBP1 in the Treacle KD cells in S5A should be in the main Figure 5 to complement transformation of constructs as in Figure 5D.

We appreciate the reviewer’s comment. To address this, we performed a quantitative analysis of PLA fluorescence signals in control and etoposide-treated cells, and the results are now presented in Supplementary Figure S8C. Additionally, as recommended, we have transferred the results of the immunocytochemistry of TOPBP1 in Treacle KD and Treacle KN cells to the main figure, now included as Figures 7D-E in the revised manuscript.